# High expression in maize pollen correlates with genetic contributions to pollen fitness as well as with coordinated transcription from neighboring transposable elements

Cedar Warman[1], Kaushik Panda[2], Zuzana Vejlupkova[1], Sam Hokin[3], Erica Unger-Wallace[4], Rex A. Cole[1], Antony M. Chettoor[3], Duo Jiang[5], Erik Vollbrecht[4,6,7], Matthew M. S. Evans[3], R. Keith Slotkin[2], John E. Fowler[1,8]*

1 Department of Botany and Plant Pathology, Oregon State University, Corvallis, Oregon, United States of America, 2 Donald Danforth Plant Science Center, St. Louis, Missouri, United States of America, 3 Department of Plant Biology, Carnegie Institution for Science, Stanford, California, United States of America, 4 Department of Genetics Development and Cell Biology, Iowa State University, Ames, Iowa, United States of America, 5 Department of Statistics, Oregon State University, Corvallis, Oregon, United States of America, 6 Bioinformatics and Computational Biology, Iowa State University, Ames, Iowa, United States of America, 7 Interdepartmental Genetics, Iowa State University, Ames, Iowa, United States of America, 8 Center for Genome Research and Biocomputing, Oregon State University, Corvallis, Oregon, United States of America

* fowlerj@science.oregonstate.edu

**Data Availability Statement:** RNA-seq generated for this study is available at the NCBI Short Read

## Abstract

In flowering plants, gene expression in the haploid male gametophyte (pollen) is essential for sperm delivery and double fertilization. Pollen also undergoes dynamic epigenetic regulation of expression from transposable elements (TEs), but how this process interacts with gene expression is not clearly understood. To explore relationships among these processes, we quantified transcript levels in four male reproductive stages of maize (tassel primordia, microspores, mature pollen, and sperm cells) via RNA-seq. We found that, in contrast with vegetative cell-limited TE expression in Arabidopsis pollen, TE transcripts in maize accumulate as early as the microspore stage and are also present in sperm cells. Intriguingly, coordinate expression was observed between highly expressed protein-coding genes and their neighboring TEs, specifically in mature pollen and sperm cells. To investigate a potential relationship between elevated gene transcript level and pollen function, we measured the fitness cost (male-specific transmission defect) of GFP-tagged coding sequence insertion mutations in over 50 genes identified as highly expressed in the pollen vegetative cell, sperm cell, or seedling (as a sporophytic control). Insertions in seedling genes or sperm cell genes (with one exception) exhibited no difference from the expected 1:1 transmission ratio. In contrast, insertions in over 20% of vegetative cell genes were associated with significant reductions in fitness, showing a positive correlation of transcript level with non-Mendelian segregation when mutant. Insertions in maize *gamete expressed2* (*Zm gex2*), the sole sperm cell gene with measured contributions to fitness, also triggered seed defects when crossed as a male, indicating a conserved role in double fertilization, given the similar phenotype previously demonstrated for the Arabidopsis ortholog *GEX2*.

Archive under accessions SRS2359914, SRS2359928, SRS2359926, SRS2359924, SRS2359918, SRS2359916, SRS2359919, SRS2359915, SRS2359925, SRS2359927, SRS2359931, SRS2359930, SRS2359921, SRS2359920, SRS2359922, SRS2359923, and SRS2359917. Previously published SRA datasets analyzed in this study are detailed in Supplemental Information S3 Table. Processed reads are also mapped to the maize B73v5 genome and viewable on the MaizeGDB genome browser at http://jbrowse.maizegdb.org

**Funding:** The work was supported by NSF Plant Genome Project grant IOS-1340050 to MME, RKS, EV and JEF. Additional support was from OSU College of Agricultural Sciences and OSU Department of Botany and Plant Pathology to JEF. The funders had no role in study design, data collection and analysis, decision to publish, or preparation of the manuscript.

**Competing interests:** The authors have declared that no competing interests exist.

Overall, our study demonstrates a developmentally programmed and coordinated transcriptional activation of TEs and genes in pollen, and further identifies maize pollen as a model in which transcriptomic data have predictive value for quantitative phenotypes.

## Author summary

In flowering plants, pollen is essential for delivering sperm cells to the egg and central cell for double fertilization, initiating the process of seed development. In plants with abundant pollen like maize, sperm cell delivery can be highly competitive. In an added layer of complexity, growing evidence indicates expression of transposable elements (TEs) is more dynamic in pollen than in other plant tissues. How these elements impact pollen function and gene regulation is not well understood. We used transcriptional profiling to generate a framework for detailed analysis of TE expression, as well as for quantitative assessment of gene function during maize pollen development. TEs are expressed early and persist, many showing coordinated activation with highly-expressed neighboring genes in the pollen vegetative cell and sperm cells. Measuring fitness costs for a set of over 50 mutations indicates a correlation between elevated transcript level and gene function in the vegetative cell. We also establish a role in fertilization for the maize *gamete expressed2* (*Zm gex2*) gene, identified based on its specific expression in sperm cells. These results highlight maize pollen as a powerful model for investigating the developmental interplay of TEs and genes, as well as for measuring fitness contributions of specific genes.

## Introduction

Sexual reproduction enables the segregation and recombination of genetic material, which increases genetic diversity in populations and contributes to the vast diversity of eukaryotes. In flowering plants, sexual reproduction requires the development of reduced, haploid gametophytes from sporophytic, diploid parents. The mature female gametophyte, the embryo sac, includes the binucleate central cell and the egg cell (reviewed in [1,2]), each of which is fertilized by a sperm cell to generate the triploid endosperm and diploid embryo, respectively. The mature male gametophyte, pollen, consists of a vegetative cell harboring two sperm cells (reviewed in [3,4]). In maize, male gametophytes arise from microspore mother cells in the tassel primordium. The transition from diploid sporophyte to haploid gametophyte occurs when these cells undergo meiosis, each resulting in four haploid microspores. Each microspore then undergoes two rounds of mitosis to produce the pollen grain, first generating the large vegetative cell and a smaller generative cell via asymmetric division, and then producing the two sperm from the generative cell. After the arrival of the pollen grain on the floral stigma, the vegetative cell transports the two sperm cells to the female gametophyte via pollen tube growth (reviewed in [5,6]). Accurate navigation of the pollen tube as it grows down the style is dependent on the architecture of the style's transmitting tract [7] and additional signaling and recognition mechanisms that are poorly understood [8]. The final stages of pollen tube growth depend on a complex interplay of signals to guide the pollen tube to the micropyle of the ovule for sperm delivery to the embryo sac (reviewed in [9]).

In maize, a pollen tube must grow up to 30 cm through the silk to reach the female gametophyte, often competing with multiple pollen tubes to eventually enter the embryo sac and release its sperm cells for fertilization (reviewed in [2,5]). Across the angiosperms, this

competitive context for pollen tube development differs, depending on the pollen population as well as sporophytic characters (reviewed in [10]). In a highly competitive environment, successful fertilization is likely enhanced by pollen tubes functioning at full capacity [11–13], as generally only the first tube to reach the micropyle is permitted to enter the female gametophyte. The mechanisms preventing entry of multiple pollen tubes, known as the polytubey block, are not well-understood, but presumably act to reduce polyspermy, which typically leads to sterile offspring [6]. In maize, mutations in the genes *MATRILINEAL/NLD/ZmPLA1* and *ZmDMP* have been linked to pollen-induced production of haploid embryos and other seed defects, which are likely associated with aberrant events at fertilization [14–17] or soon after [18]. Thus, many mechanisms associated with both pollen tube growth and fertilization remain enigmatic.

Given their specialized biological functions and well-defined developmental stages, gametophytes are prime targets for transcriptome analysis. Initial studies of plant gametophytic transcriptomes in Arabidopsis pollen [19,20] and embryo sacs [21,22] described a limited and specialized set of transcripts and identified numerous candidate genes for gametophytic function. In maize, the first RNA-seq study of male and female gametophyte transcriptomes (mature pollen and embryo sacs) similarly identified subsets of developmentally specific genes, with pollen showing the most specialized transcriptome relative to other tissues assessed [23]. More recently, RNA-seq has been carried out on additional stages of maize reproductive development, including pre-meiotic and meiotic anther cells [24–26], as well as sperm cells, egg cells, and early stages of zygotic development [27].

Gametophytic tissues are known to show dynamic expression of transposable elements (TEs). In Arabidopsis, global TE expression is derepressed at the late stages of pollen development, occurring in the pollen vegetative nucleus only after pollen mitosis II [28]. The pollen vegetative nucleus undergoes a programmed loss of heterochromatin, resulting in TE activation, TE transposition and subsequent increased RNA-directed DNA methylation [28–32]. A variety of functions have been ascribed to this male gametophytic "developmental relaxation of TE silencing" (DRTS) event [33], including the generation of TE small interfering RNAs that are mobilized to the sperm cells [34], and control of imprinted gene expression after fertilization [35]. However, the dynamics of TE expression during gametophytic development in a transposable element-rich species such as maize have not been investigated.

To provide a more full description of transcriptome dynamics across maize male reproductive development, including TE transcriptional activity, we generated RNA-seq datasets from tassel primordia, microspores, mature pollen, and isolated sperm cells. Using these data, we describe differential expression patterns of genes and TEs across these stages, uncovering a coordinated regulation of TEs and their neighboring genes in pollen grains. Then, within a framework provided by the transcriptome data, we conducted a functional validation of highly expressed genes by testing over fifty insertional mutations for male-specific fitness effects. Finally, the same transcriptome data guided the discovery of mutant alleles in the sperm cell-enriched *gex2*, which induces seed development defects when present in the pollen parent, implying a role in fertilization.

## Results

### Experimental design and gene expression during maize male reproductive development

RNA-seq was performed on four tissues representing integral stages in maize male gametophyte development: immature tassel primordia (TP), isolated unicellular microspores (MS), mature pollen (MP), and isolated sperm cells (SC) (Fig 1A). Techniques were developed to

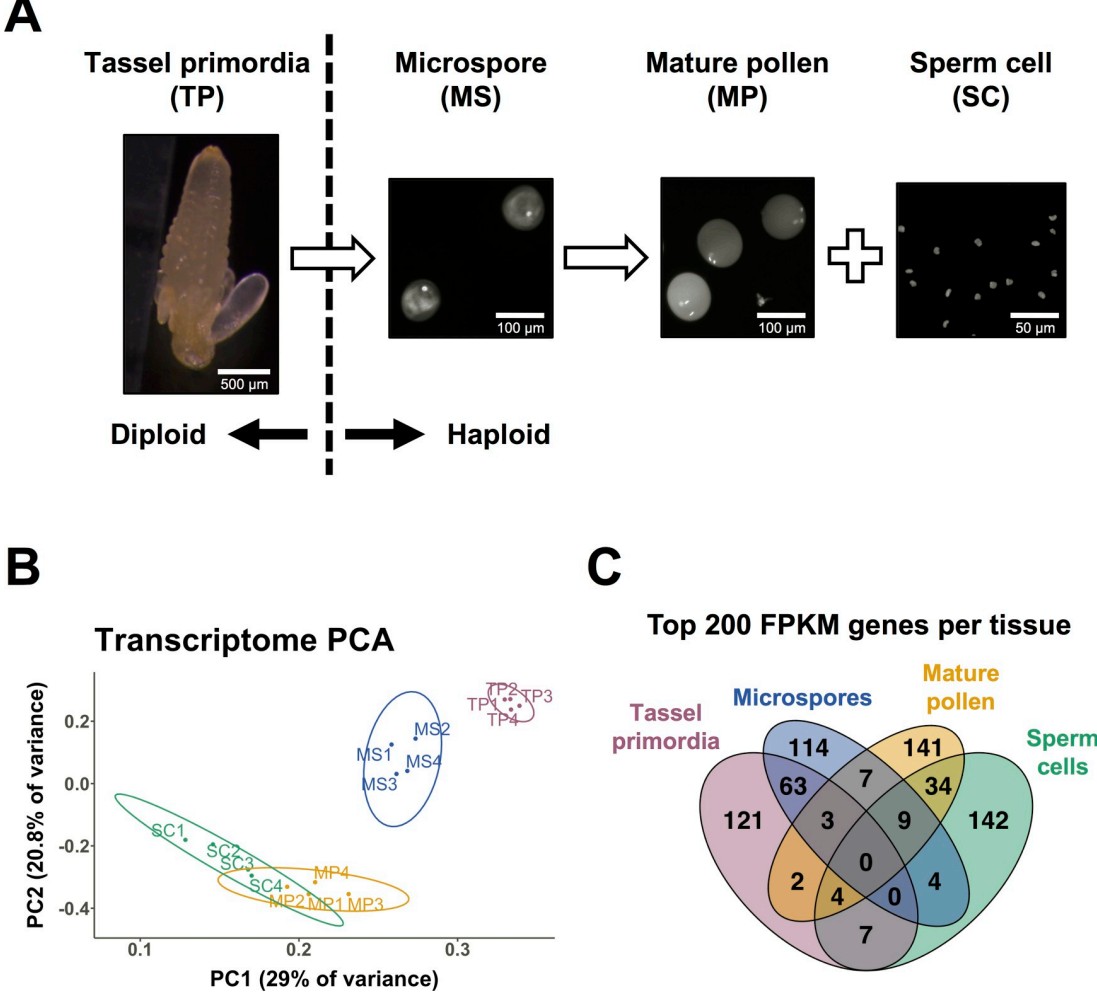

**Fig 1. Experimental design and characteristics of maize male reproductive transcriptomes.** (A) mRNA was isolated from four developmental stages of maize male reproductive development, with four biological replicates for each: pre-meiotic tassel primordia (TP), post-meiotic, unicellular microspores (MS), mature pollen (MP), and sperm cells (SC). A single biological replicate of mRNA from the bicellular stage of pollen development was also isolated and sequenced (MS-B, not shown here). Nuclei were stained with either DAPI or Dyecycle green. (B) Principal component analysis of genic transcriptomic data generated by this study, showing the 2 major components (explaining 49.8% of the variance) on x- and y-axis. The four biological replicates of each of the four sequenced tissues clustered with other replicates from the same tissue. TP and MS were clearly separated in principal component space, whereas SC and MP samples displayed less separation from each other. (C) High expression levels are associated with developmental specificity: approximately 2/3 of the genes associated with the highest FPKM values in each of the four sample types are highly expressed in only that sample type.

efficiently isolate RNA from TP, MS, and SC (see Methods). RNA was extracted from the inbred maize line B73, with four biological replicates for each tissue. In addition, a single RNA replicate was isolated for the bicellular stage of pollen development (MS-B). Libraries were sequenced using Illumina sequencing (100 bp paired-end reads) and mapped to the B73 AGPv4 reference genome [36]. Principal Component Analysis (PCA) showed samples from each tissue clustering together along PC1 and PC2, which together explained 49.8% of the variance between samples (Fig 1B). One sample, SC1, had significant levels of ribosomal RNA (rRNA) contamination, as well as the fewest number of mapped reads (approximately 1 million). However, to maintain a balanced experimental design with a consistent false discovery rate (FDR), we chose to include SC1 in our analysis of gene expression patterns.

Differential gene expression was defined in two ways: in the first, gene expression in later developmental stages was compared to the premeiotic, diploid tassel primordia (TP vs MS, TP vs MP, and TP vs SC); in the second, gene expression was compared between all adjacent developmental stages (TP vs MS, MS vs MP, MS vs SC, MP vs SC) (S2 and S11 Tables). Enriched GO terms highlighted the differences in gene expression among developmental stages and suggested consistency with the established functions of each tissue [19,23,27]. GO terms in MS were consistent with a post-meiotic tissue still at an early stage of development, with terms related to protein synthesis and transport, morphogenesis, and reproduction showing enrichment. MP showed more specific enriched GO terms, including those related to pollen tube growth, signaling, and actin filament-based movement. SC shared many GO terms with MP when compared to MS, but was uniquely enriched for GO terms related to epigenetic regulation of gene expression, such as histone H3K9 demethylation and gene silencing by RNA, potential mechanisms involved in differential regulation of TEs.

Comparison of the most highly expressed genes from all four sample types showed that, generally, such transcripts were highly enriched at a single developmental stage (Fig 1C). Notably, no single gene was highly expressed in all four tissues, and fewer than twenty were highly expressed in three out of the four stages assessed. These data suggest a simple hypothesis in which the expression level for protein-coding genes reflects, at some level, functional importance, i.e., a high expression level at a specific developmental stage implies an increased contribution by the gene's encoded function at that particular stage. Alternatively, or in addition, high expression could be reflective of regulatory mechanisms specific to each stage, each primarily influencing specific subsets of genes. We were interested to explore the possibility of some regulatory linkage between those genes and TEs highly expressed in the male gametophyte.

## A subset of transposable elements in the maize genome show developmentally dynamic expression

To obtain a baseline view of TE expression throughout maize development, our RNA-seq data for maize male reproductive development (samples with asterisks, S1 Fig) was combined with publicly available datasets from nine-day old above-ground seedlings, juvenile leaves, ovules, another set of independently isolated sperm cells, and three independent studies of pollen RNA-seq [23,27,37,38] & SRP067853). The complete list of samples, their sequencing statistics, references and data availability can be found in S3 Table. All of the raw data were remapped using the same parameters (see Methods). Principal component analysis demonstrates that replicates of the same tissue and growth state typically group together (S1 Fig).

We aimed to identify the set of dynamically expressed TEs within the tissues sampled, and thus calculated expression levels for each individual TE in the genome located more than 2 kb away from annotated non-TE genes. Our rationale was to avoid false positive signals of TE expression due to a TE residing within a gene, and to minimize the influence of read-through transcription from a nearby gene, which could not be distinguished from TE-initiated transcription. Because we concentrated on individual elements, and not TE families, the majority of annotated TEs were not assessed in this analysis (55%; Fig 2A 'not covered'), either because no expression was detected in any dataset, or because their sequence lacks the polymorphisms necessary for mapping reads to a specific TE. To relate TE expression comparatively across development, we used seedling tissue as a baseline against which other tissues were measured. Seedling was chosen for several reasons: it is not a reproductive tissue, it has low to average levels of TE expression, and a large number of TEs show no evidence of expression in this tissue (S2 Fig).

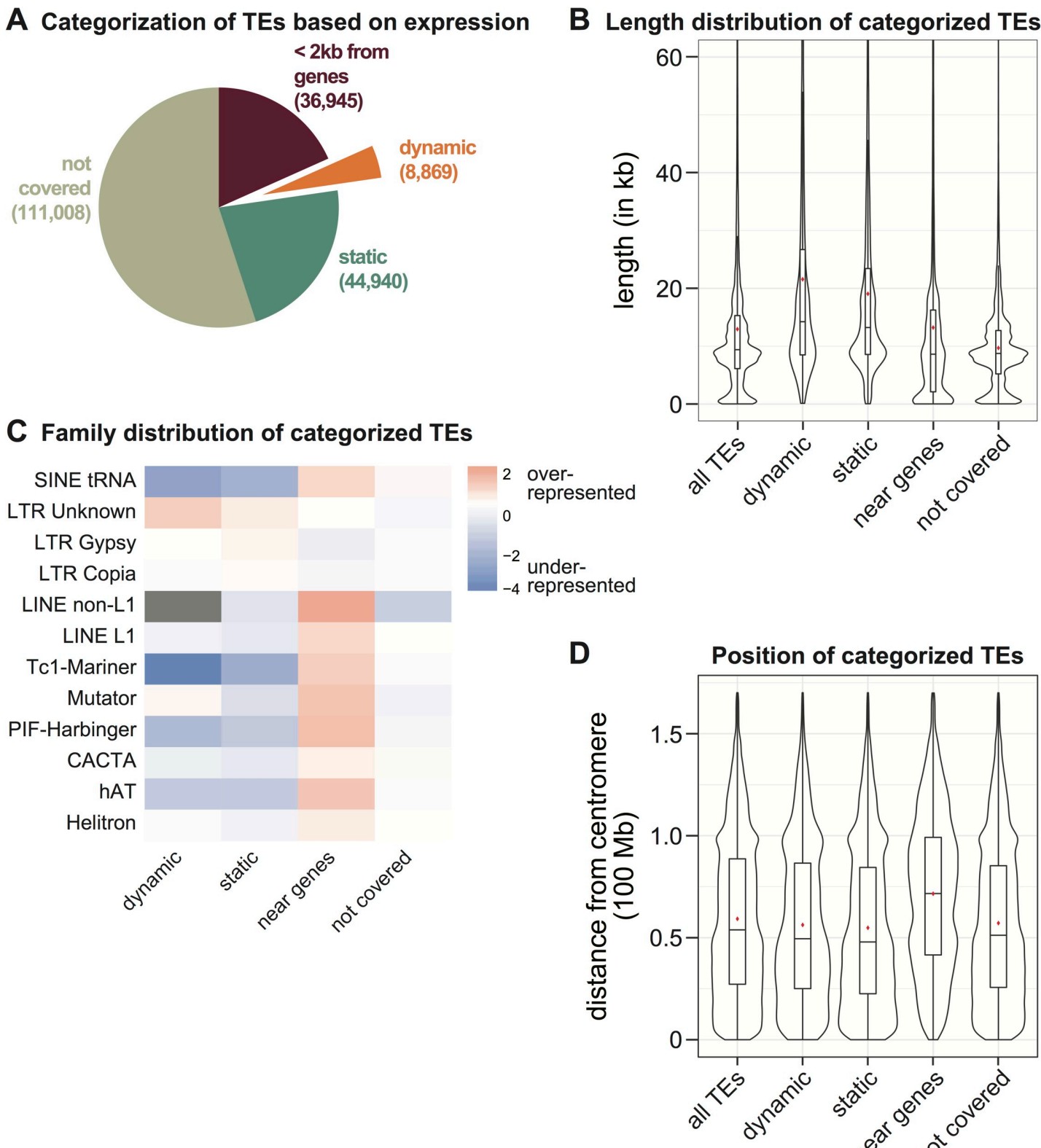

**Fig 2. Characterization of developmentally dynamic transcription from transposable elements (TEs).** (A) Distribution of different categories of TEs based on their expression. Number of TEs are in parentheses. (B) Length of TEs in the different TE categories from part A. The violin plots around the box show the kernel probability density of the data. The box represents lower and upper quartile, the line is the median, and the whiskers represent 10–90% range. Red asterisk denotes the mean. **(C)**

Observed / expected Log2 ratios of TE family proportions in the different TE categories from part A. Grey indicates no data. **(D)** Distance from the annotated centromere for different TE categories from panel A.

Apart from the 18.3% of annotated TEs that are near genes and analyzed separately (see below), we calculated the number of TEs with statistically significant expression differences in each tissue compared to the seedling reference. This identified the subset of TEs that are developmentally dynamic, meaning that they show differential expression in at least one tissue in our dataset compared to the seedling reference. Only 4.4% of all maize annotated TEs are developmentally dynamic, whereas 22.2% of TEs have detectable expression but do not change in our dataset and therefore are developmentally 'static' (Fig 2A). Each TE category was interrogated for feature overrepresentation. Both dynamic and static TEs are longer than the genome average, and longer than the sets of TEs 'not covered' or 'near genes' (Fig 2B). To determine if one long family of TEs was contributing this difference, we performed this analysis for each TE superfamily and found that for dynamic TEs, this observation is not specific to one TE type (S3 Fig). The finding that expressed TEs as a group are longer correlates with Arabidopsis data where longer TE transcripts are overrepresented and differentially regulated when epigenetic repression is lost [39].

Expressed TEs show an under-representation for DNA transposon and SINE families, which are mainly within the 'near genes' set (Fig 2C). In contrast, the 'LTR unknown' TE annotation is over-represented in the dynamic TE set (Fig 2C). Since some LTR retrotransposons are enriched in the pericentromere [40], we tested if the dynamic TE set is enriched in the pericentromere compared to the genome average, but did not detect any correlation (Fig 2D). Therefore, we conclude that expressed TEs are generally longer elements, and the subset of developmentally dynamic TEs are enriched for uncharacterized LTR retrotransposons located throughout the genome.

## Transposable element transcript levels are up-regulated in the post-meiotic male reproductive lineage

From the developmentally dynamic TE set, we calculated the number of differentially expressed TEs in each tissue/stage compared to the seedling reference. In some tissues, such as tassel primordia and ovules, we observed a similar number of TEs up-regulated and down-regulated (Fig 3A), demonstrating that while there are shifts in which TEs are expressed, a genome-scale change in TE expression does not occur. In other tissues, such as juvenile leaves, there is a skew towards increased TE expression. The largest TE up-regulation occurs in the tissues of the male reproductive lineage, including unicellular and bicellular microspores, mature pollen and isolated sperm cells (Fig 3A). Our data confirm the recent finding that the tissue with the largest number of TE families activated is mature pollen [41]. The number of up-regulated TEs compared to down-regulated TEs in these tissues suggest that there is a genome-wide activation of TE expression, similar to the DRTS event that occurs in Arabidopsis pollen [28,33]. One important distinction is that TE expression is present in maize sperm cells (Fig 3A), whereas it is not detected in Arabidopsis sperm cells [28]. To verify this finding, we compared our sperm cell RNA-seq data to an independent maize sperm cell dataset [27]. We found that TEs are also significantly expressed in this independent dataset, and 70% of those expressed TEs are also detected in our dataset (p<0.001) (Fig 3B). This shared set of 810 sperm cell-expressed TEs (38% of those detected in our dataset), supports the conclusion that significant expression of TEs occurs in maize sperm cells. Of the sperm cell-expressed TEs, 36% were not observable in total pollen, but rather required the isolation and enrichment of

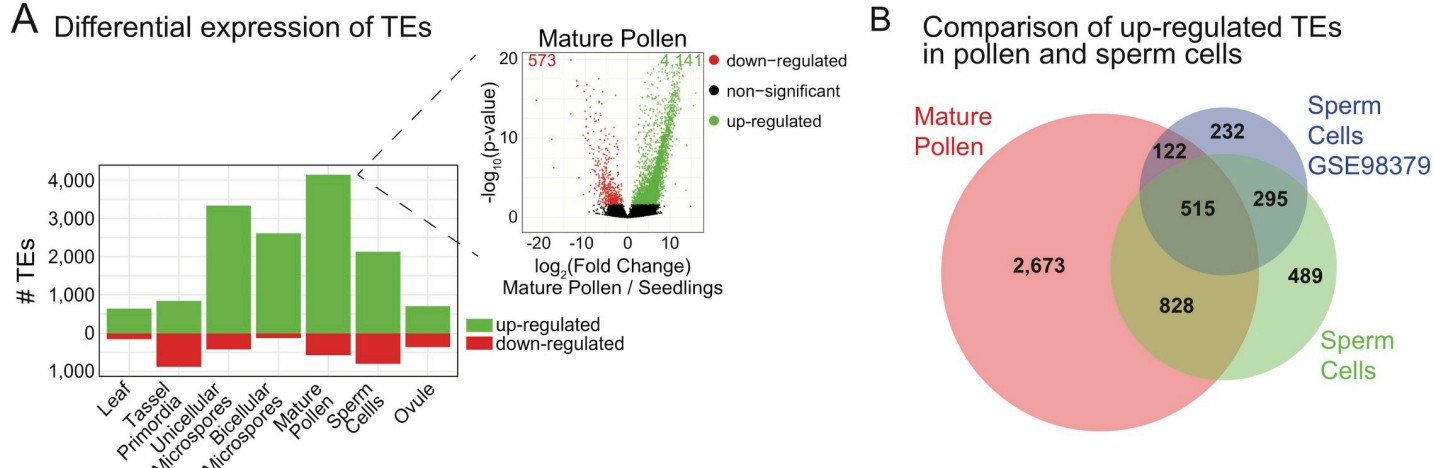

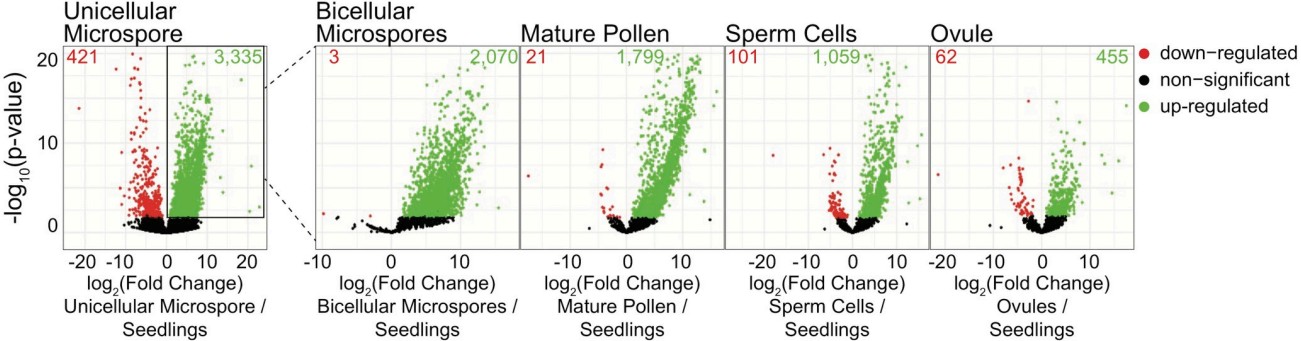

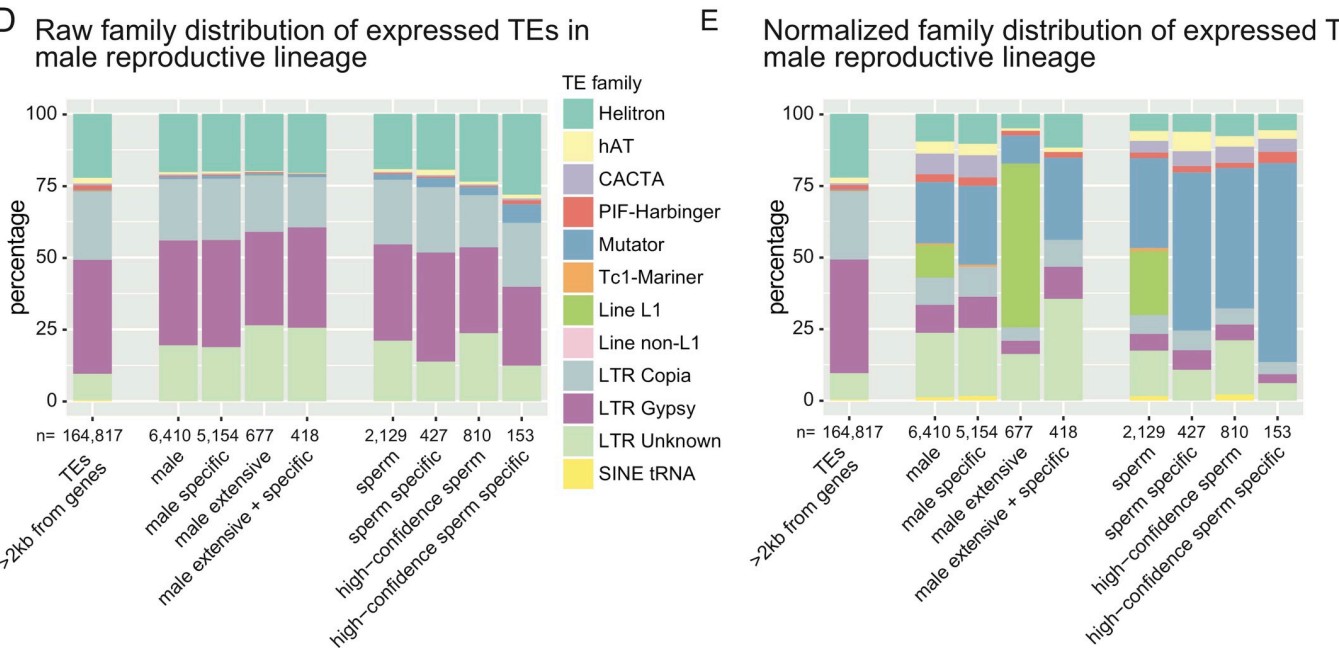

**Fig 3. High TE expression in the maize male gametophyte lineage.** (A) Number of differentially expressed TEs in seven tissues compared to seedlings. The inset volcano plot shows for mature pollen how differentially expressed TEs were identified. Green and red numbers within the volcano plot indicate how many TEs were statistically up- or down-regulated, respectively. (B) Number of up-regulated TEs in mature pollen compared to isolated sperm cells from this study and a previously published distinct isolation and sequencing of sperm cell mRNA. (C) Starting with TEs differentially up-regulated in unicellular microspores (boxed, far left volcano plot), we determined how many of these same TEs are expressed at other developmental time points. (D) Raw distribution of expressed TE family annotations. 'Male'

refers to the set of TEs expressed in any male lineage dataset (MS, MS-B, MP, SC). 'Male specific' are TEs expressed in only the male lineage (not other tissues / timepoints). 'Male extensive' TEs are expressed in all of the male lineage datasets, and 'male extensive + specific' refers to TEs expressed all male lineage datasets and not other tissues / timepoints. 'High-confidence sperm' refers to TEs identified as expressed in both analyzed sperm cell datasets from part B. **(E)** Expressed TE family annotations normalized to the genome-wide TE distribution of TEs >2 kb from genes. Categories are the same as part D.

sperm cells for detection (Fig 3B). Overall, we detect 157 TEs expressed in both sperm cell datasets that are not expressed throughout development, but specifically in the sperm cells (sperm-cell exclusive).

A second notable difference between maize and Arabidopsis is the activation of TE expression early in the male gametophytic phase of maize. A genome-wide increase in TE transcript levels is detected at the earliest post-meiotic stage tested, the microspore, in contrast to low TE expression in the sporophytic tassel primordia (Fig 3A). Arabidopsis TE expression occurs only late in pollen development, after pollen mitosis I when the vegetative cell is generated [28]. To determine if TEs were indeed activated early in maize male reproductive development, we asked if the same TEs that we identified as expressed in the unicellular microspore remain active throughout the male reproductive lineage. We used the set of differentially expressed up-regulated TEs in unicellular microspores (3,335) and found that 62% are still expressed in bicellular microspores and 54% in mature pollen (Fig 3C), demonstrating that once TEs are activated early in development, expression and/or steady-state mRNA frequently remains through pollen maturation. Only some of these male-lineage expressed TEs continue to be expressed in sperm cells (32%), raising the possibility that many TEs with active expression in the early gametophytic stages are under negative/repressive regulation in the gametes. This large-scale developmental activation is potentially limited to the male lineage, as ovules express relatively few TEs (Fig 3A) and only 14% of the male lineage-expressed elements (Fig 3C). Together, our data demonstrate conserved activation of TE expression in the male gametophytes of maize and Arabidopsis, with key differences such as the developmental timing and localization of TE expression in the gamete cells.

We determined what types of TEs activate in the male reproductive lineage and sperm cells and compared these to the whole-genome distribution of TEs analyzed. Overall, both male lineage-expressed TEs and sperm cell-expressed TEs reflect the genome-wide TE distribution (Fig 3D). This suggests that TE family type does not have a determining role in the developmental regulation of TE expression. One notable exception is the enrichment of *Mutator* family TE expression in sperm cells (Fig 3D). When normalized for genome-wide TE distribution, *Mutator* element expression is highly enriched across the male lineage, including in sperm cells (Fig 3E). The expression of some *Mutator* TEs in sperm cells is both high confidence (present in both sperm cell datasets) and specific to only that tissue (high confidence sperm cell specific, Fig 3E). LINE L1 elements are also expressed throughout the male lineage and sperm cells, but their expression is general and not specific to these cell types (Fig 3E). Our data demonstrate that there is a general (TE family-independent) activation of TE expression in the male reproductive lineage, with one observable bias towards *Mutator* family expression in both the male lineage and sperm cells.

## Mature pollen and sperm cells display coexpression of highly expressed genes and their neighboring TEs

To determine if TEs have an effect on neighboring gene expression, or vice versa, we next analyzed the set of 36,945 assayable TEs within 2kb of genes (Fig 2A). We calculated the absolute expression level of each genic isoform and categorized them into 100 bins of expression levels for each developmental stage (Fig 4). We found no significant relationship between highly expressed genes and the number of up- or down-regulated TEs in tassel primordia or

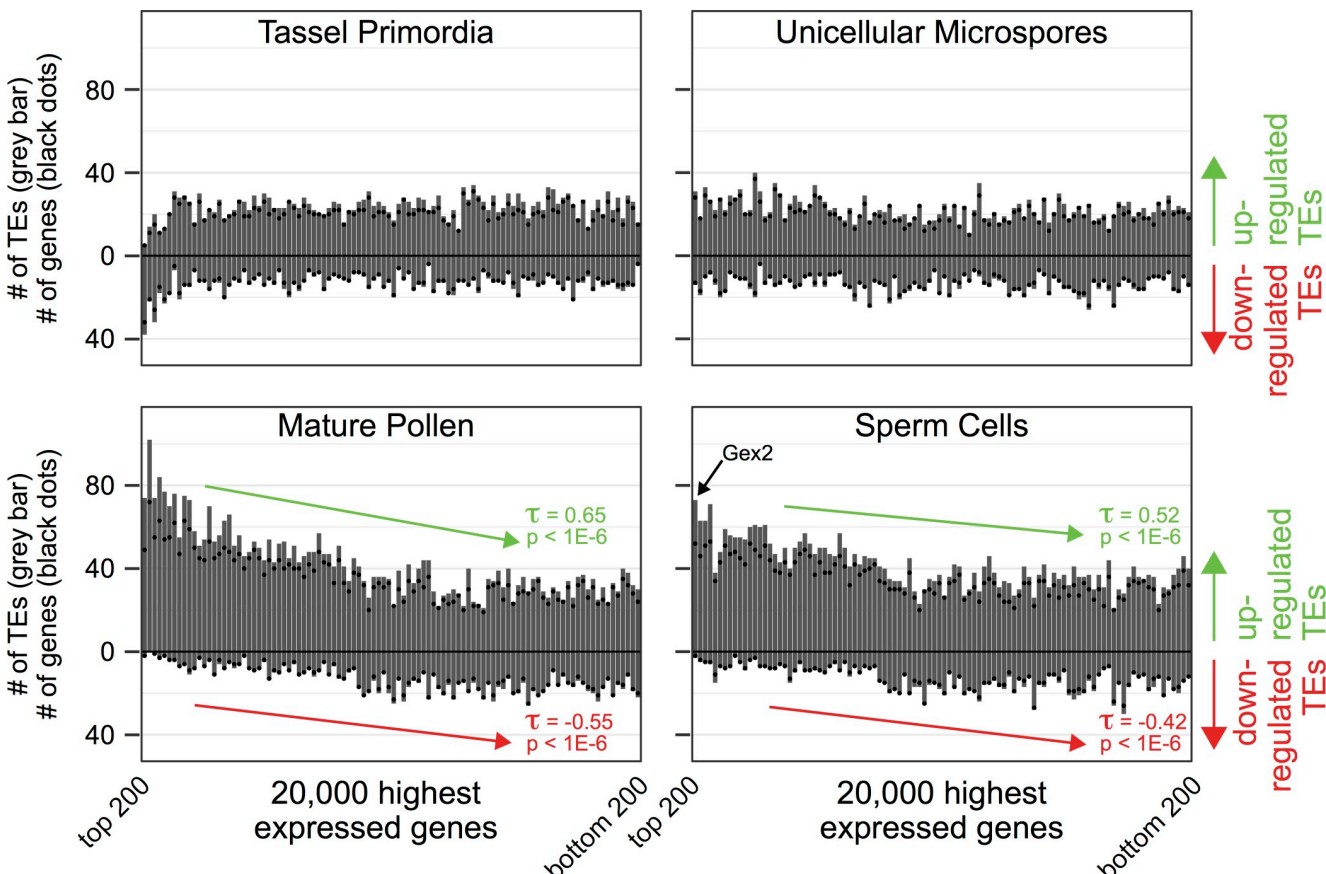

**Fig 4. Co-regulation of TE and gene expression in the male gametophyte.** For each tissue type, the top 20,000 most highly expressed genes are distributed along the X-axis in bins of 200, with the most highly expressed bin on the far left. For each bin the number of up- and down-regulated TEs near (<2kb) that bin's genes is then summed on the Y-axis (shown in grey bar). For each bin, the number of genes with at least 1 up- or down-regulated TE within 2kb is displayed as black dots. In unicellular microspores (top right) there is little correlation, whereas in mature pollen and sperm cells (bottom panels) the most highly expressed genes are near primarily up-regulated TEs. To check if the perceived correlations are statistically significant, we performed Kendall's Tau rank correlation tests and found significant correlations (p < 1E-6) only for mature pollen and sperm cells for both number of TEs and number of genes. For mature pollen, gene expression has a positive correlation with up-regulated TEs (#TEs: $\tau$ = 0.65, #genes: $\tau$ = 0.64) and a negative correlation with down-regulated TEs (#TEs: $\tau$ = -0.55, #genes: $\tau$ = -0.55). Similarly for sperm cells, gene expression has a positive correlation with up-regulated TEs (#TEs: $\tau$ = 0.52, #genes: $\tau$ = 0.52) and a negative correlation with down-regulated TEs (#TEs: $\tau$ = -0.42, #genes: $\tau$ = -0.44). Only TE (not genes) rank correlation coefficients ($\tau$) are displayed in the figure. The bin location of *gex2* (see Fig 7) is annotated in the sperm cell data.

microspores (grey bars, top row, Fig 4). In contrast, in both mature pollen and isolated sperm cells there is a statistically significant (p < 1E-6) positive association between highly expressed genes and the number of up-regulated TEs within 2kb of those genes (grey bars, bottom row, Fig 4). Similarly, there is a negative correlation (p < 1E-6) between high gene expression and the number of down-regulated TEs in the same samples (grey bars, bottom row, Fig 4). This relationship is not due to the fact that pollen or sperm cell-expressed genes are more likely to be located nearby a TE (S4A Fig), nor due to sample contamination between these two datasets (S4B Fig). To determine if our analysis is biased by the presence of multiple TEs close to just a few highly expressed genes, we also counted the number of genes with at least one differentially expressed TE within 2kb (black dots, Fig 4 and S4 Fig). We found that a similar number of genes were adjacent to differentially expressed TEs (compared grey bars to black dots, Fig 4 and S4 Fig), demonstrating that a small number of genes was not biasing our dataset. We conclude that specifically in the mature male gametophyte the most highly expressed genes tend to be near actively expressing TEs. However, it remains unclear whether gene expression is

influencing TE expression, or TE expression is affecting gene regulation, or alternatively, some global regulatory mechanism is influencing both.

## Large-scale insertional mutagenesis supports a relationship between transcript level and fitness contribution for vegetative cell-expressed genes

Using the quantitative framework provided by our transcriptome dataset, we next tested the simple hypothesis that highly expressed genes contribute to male gametophytic function–i.e, to reproductive success (pollen fitness). The functional validation approach we used relied on a large, sequence-indexed collection of green fluorescent protein (GFP)-marked transposable element (*Ds-GFP*) insertion mutants [42], enabling assessment of the effects of mutations in select genes (Fig 5). We focused on expression data from the MP and SC stages, as these have

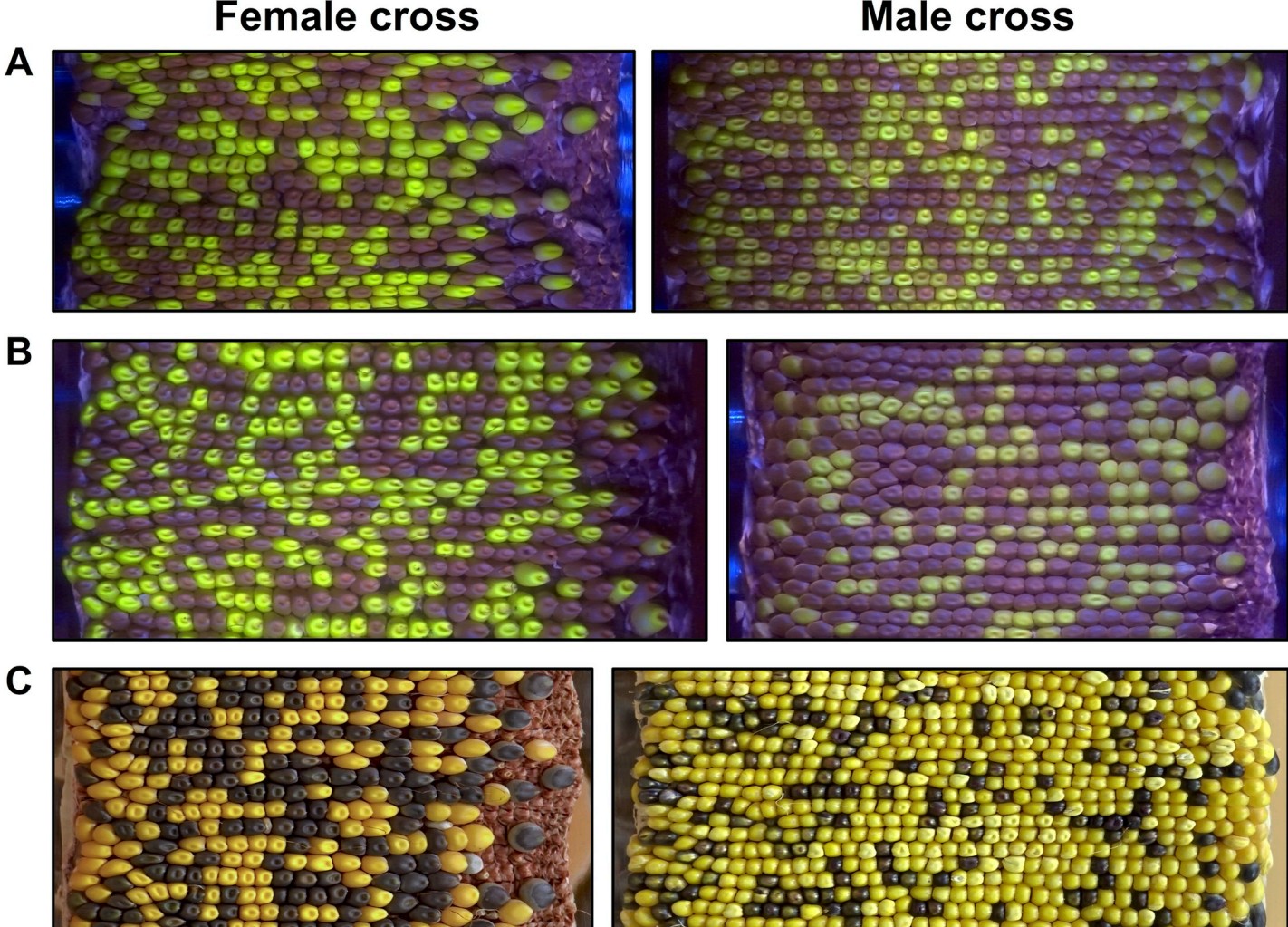

**Fig 5. Large-scale tracking of seed marker transmission frequencies was accomplished by generating ear projections with a custom built rotational scanner. (A)** When crossed either through the male or the female, *Ds-GFP* mutant allele *tdsgR107C12* (in gene Zm00001d012382), marked by green fluorescent seeds, shows 1:1 Mendelian inheritance (50% transmission of the GFP seed marker). Images captured in blue light with an orange filter. **(B)** Mutant alleles in other genes, such as *tdsgR102H01* (Zm00001d037695), showed non-Mendelian segregation when crossed through the male (37.5% GFP transmission). Segregation through the female remained Mendelian, indicating a male-specific transmission defect. **(C)** For some mutant alleles (~10% of lines in this study), the anthocyanin transgene *C1* was tightly linked to the insertion mutant. In these cases, seeds carrying a mutant allele of a gene of interest could be tracked by their purple color. Here, insertion *tdsgR96C12* (Zm00001d015901) shows a strong male-specific transmission defect (24.8% *C1* transmission through the male). Images captured in full spectrum visible light.

distinctive cell fates and roles in reproduction: the vegetative cell generates the pollen tube for competitive delivery of gametes, and the sperm cells accomplish double fertilization. Expression data from seedlings [23] was used to design a sporophytic control. Highly expressed genes, operationally defined as in the top 20% for a tissue by FPKM, were grouped into three mutually exclusive classes: Seedling, Sperm Cell, and Vegetative Cell. The seedling group also excluded any gene highly expressed in either MP or SC. Due to the significant overlap among genes highly expressed in both MP and SC, we compared expression values to assign each of these genes to a single class. Vegetative Cell genes were not only highly expressed in MP, but were also associated with an FPKM greater in MP than in SC, and vice versa for Sperm Cell genes (S5 Table). All genes in these classes were then cross-referenced with *Ds-GFP* insertion locations to identify potential mutant alleles for study, restricting the search to insertions in coding sequence (CDS), as these were rationalized as most likely to generate loss-of-function effects. Finally, to insure our results were as generalizable as possible, each class list was randomized to identify the specific subset of *Ds-GFP* lines for study. Insertion locations were verified by PCR for 64 of 83 alleles obtained (S6 Table) (see Methods), of which 56, representing mutations in 52 genes, generated sufficient transmission data to include in our final analysis.

Mendelian inheritance predicts 50% transmission of mutant and wild-type alleles when a heterozygous mutant is outcrossed to a wild-type plant. However, a mutation that alters the function of a gene expressed during the haploid gametophytic phase can result in a reduced transmission rate if that gene contributes to the fitness of the male gametophyte–i.e., to its ability to succeed in the highly competitive process of pollen tube growth, given that 50% of the pollen population will be wild-type for the same gene. Thus, reduced transmission of a mutant through the male (a male transmission defect) provides not only evidence for gene function in the gametophyte, but also a measure of the mutated gene's contribution to fitness. Transmission rates through the female serve as a control, as 50% transmission through the female would confirm both a single *Ds-GFP* insertion in the genome and male-specificity for any defect identified. To measure the fitness cost of each *Ds-GFP* insertion, heterozygous mutant plants were reciprocally outcrossed with a heavy pollen load to a wild-type plant, maximizing pollen competition within each silk. Transmission rates were then quantified by assessing the ratio of the non-mutant to mutant progeny using a novel scanning system and image analysis pipeline (Fig 5) (see Methods) [43]. Mutant alleles were tracked using linked endosperm markers: either the GFP encoded by the inserted transposable element (Fig 5A and 5B), or, in ~10% of the lines, a tightly linked *C1*+ anthocyanin transgene (present due to the initial *Ds-GFP* generation protocol) (Fig 5C, S7 Table).

Transmission rates for all groups were tested through quasi-likelihood tests on generalized linear models with a logit link function for binomial counts (see Methods, S8 Table). When crossed through the female, no genes showed significant differences from Mendelian inheritance (Fig 6A). When crossed through the male, no genes with insertion alleles in the Seedling category (n = 10) showed evidence of abnormal transmission rates (Fig 6B). Most Sperm Cell genes (n = 10, 90%) showed no statistically significant transmission defects, with one notable exception (two independent alleles of the *gex2* gene, described in detail below) (Fig 6C). However, among Vegetative Cell genes tested (n = 32), a larger proportion of insertion alleles (7 out of 32 or 21.9%) showed significant male transmission defects (quasi-likelihood test, adjusted p-value threshold < 0.05) (Fig 6D). The proportions of genes with transmission defects in the three classes were not significantly different by Fisher's exact test (Seedling vs Sperm Cell p-value = 0.500, Seedling vs Vegetative Cell p-value = 0.125, Vegetative Cell vs Sperm Cell p-value = 0.374), likely due to the small number of mutations assessed in the Seedling and Sperm Cell classes. For the insertion alleles tested, a summary description of genes showing non-Mendelian inheritance can be found in Table 1, whereas a description of those showing Mendelian inheritance can be found in Table 2.

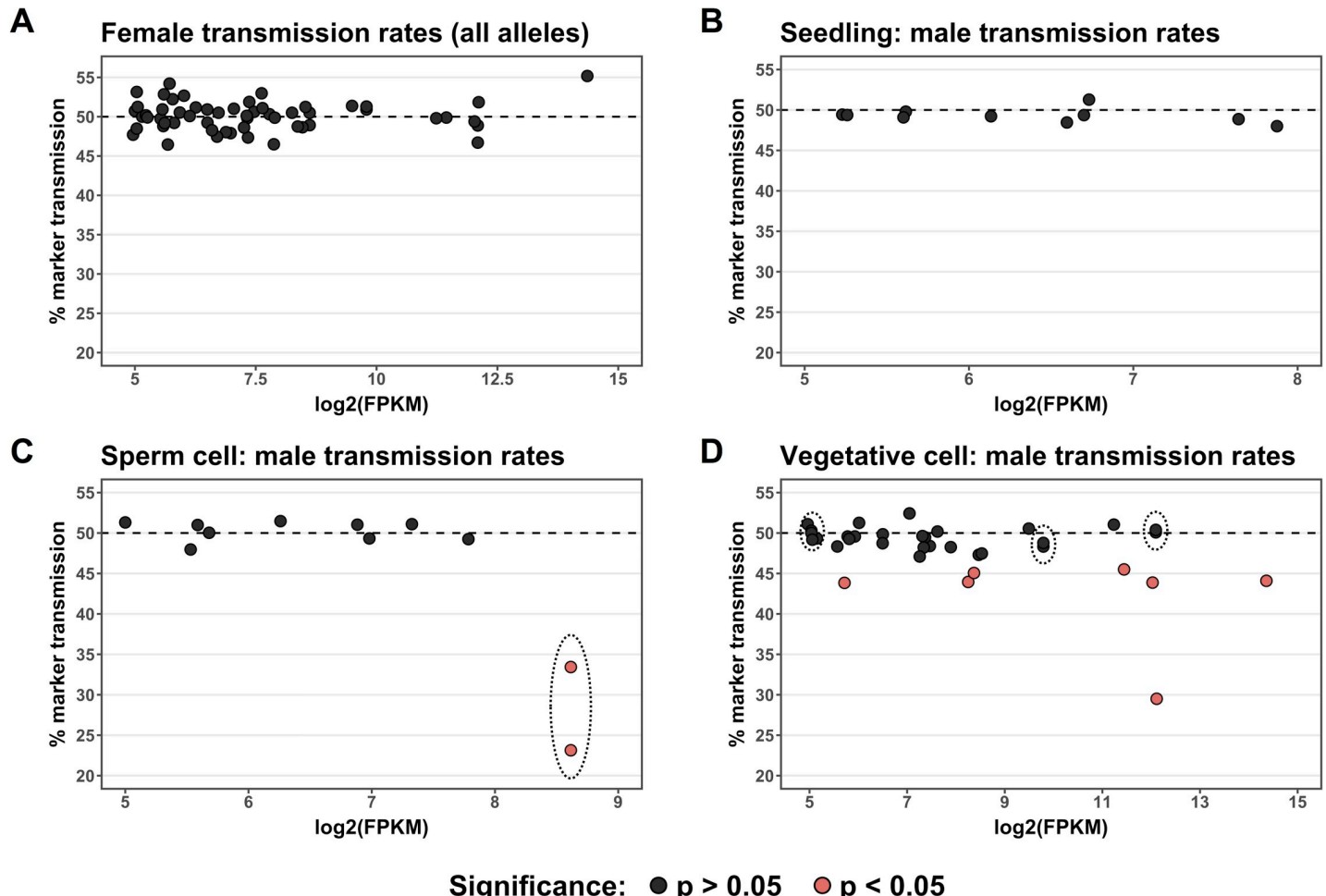

**Fig 6. Functional validation of highly-expressed gametophyte genes by quantification of transmission rates in *Ds-GFP* insertional mutants.** Alleles with CDS insertions were tested for differences from Mendelian inheritance using a quasi-likelihood test, with p-values corrected for multiple testing using the Benjamini-Hochberg procedure; alleles with quasi-likelihood adjusted p-value < 0.05 are represented in pink. Alleles are plotted by the log2(FPKM) of their respective gene according to that gene's expression class (Seedling, Vegetative Cell, or Sperm Cell). Insertion alleles distributed among the classes as follows: Vegetative Cell, 35 alleles; Sperm Cell, 11 alleles; Seedling, 10 alleles. The number of seeds categorized for each allele ranged from 1,522 to 5,219, with an average of 2,807. Genes represented by two independent insertion alleles are enclosed by dotted lines. (A) Transmission rates of 56 mutant allele seed markers for heterozygous Ds-GFP mutant plants crossed through the female. (B) Transmission rates for alleles in the negative control Seedling class when crossed through the male. (C) For genes belonging to the Sperm Cell class, one out of ten (10%) was associated with significant non-Mendelian inheritance. The single gene with a male transmission defect in this group (gex2) showed a strong defect for both of the independent alleles tested. (D) An increased proportion of the genes in the Vegetative Cell class were associated with significant non-Mendelian inheritance when mutant (7/32 genes, 21.9%). In this class, an increase in log2(FPKM) was significantly associated with a decrease in marker transmission (linear regression, p = 0.0120).

The majority of transmission defects in the Vegetative Cell class genes (six of the seven with significant effects) were mild, at approximately 45% transmission, with only one reducing transmission by a moderate amount, to ~30%. Notably, six of the genes associated with significant defects were measured at a $\log_2$(FPKM) > 8 (i.e., in the top 5% of Vegetative Cell genes by FPKM). Given that twelve genes above this threshold were tested, these most highly expressed Vegetative Cell genes were significantly more likely to be associated with non-Mendelian transmission (6 out of 12) than the group of Vegetative Cell genes below this expression threshold (1 out of 20) (Fisher's exact test, p-value = 0.00572). Consistent with this observation, an increase in $\log_2$(FPKM) was associated with both reduced transmission rate and an increase in -$\log_{10}$(p-value) (linear regression, p-value = 0.0120, 0.0255, respectively). Thus, our

**Table 1.  Characteristics of genes showing non-Mendelian inheritance.**

| Category | Gene designation (v3) | Gene designation (v4) | Gene Type | Ds-GFP allele | Male transmission rate | Adjusted p-value | Best BLAST Hit, A. thaliana | Predicted Function (B73v4 Gramene) | Cellular process (inferred) |
|---|---|---|---|---|---|---|---|---|---|
| Vegetative Cell | GRMZM2G359879 | Zm00001d028437 | singleton | tdsgR04A02 | 43.84% | 3.29E-04 | AT3G61050 | Calcium-dependent lipid-binding (CaLB domain) family protein | Cell signaling |
| Vegetative Cell | GRMZM2G350802 | Zm00001d037695 | singleton | tdsgR102H01 | 45.50% | 3.17E-02 | AT1G52080 | Actin binding protein family | Cytoskeleton |
| Vegetative Cell | GRMZM2G039583 | Zm00001d022250 | singleton | tdsgR33F03 | 43.95% | 1.38E-04 | AT2G02370 | SNARE associated Golgi protein family | Vesicle trafficking |
| Vegetative Cell | GRMZM2G012328 | Zm00001d003431 | syntelog | tdsgR49F11 | 43.87% | 1.38E-04 | AT3G05610 | Pectinesterase 5 | Cell wall modification |
| Vegetative Cell | GRMZM2G135570 | Zm00001d014731 | singleton | tdsgR67C09 | 44.09% | 1.06E-03 | AT2G29960 | Peptidyl-prolyl cis-trans isomerase CYP20-1 | Protein folding |
| Vegetative Cell | GRMZM2G153987 | Zm00001d014782 | singleton | tdsgR92F08 | 45.06% | 1.41E-02 | AT1G19940 | Endoglucanase 2 | Cell wall modification |
| Vegetative Cell | GRMZM2G082517 | Zm00001d015901 | singleton | tdsgR96C12 | 29.51% | 0.00E+00 | AT2G24450 | Fasciclin-like arabinogalactan protein 3 | Cell wall modification |
| Sperm Cell | GRMZM2G036832 | Zm00001d005781 | singleton | tdsgR82A03 | 33.43% | 4.15E-14 | AT5G49150 | Protein GAMETE EXPRESSED 2 | Fertilization |
| Sperm Cell | GRMZM2G036832 | Zm00001d005781 | singleton | tdsgR84A12 | 23.14% | 0.00E+00 | AT5G49150 | Protein GAMETE EXPRESSED 2 | Fertilization |

data indicate that transcript level in the Vegetative Cell does provide some limited predictive power for identifying gene-specific contributions to male gametophytic fitness (adjusted $R^2$ = 0.151, 0.116, respectively). Vegetative Cell genes associated with non-Mendelian inheritance had a range of predicted cellular functions, including cell wall modification, cell signaling, protein folding, vesicle trafficking, and actin binding (Table 1).

To ensure the experimental design was robust, we examined two potential confounding variables: the presence of the *wx1-m7*::*Ac* allele in a subset of lines tested and the potential for epigenetic silencing of GFP transgenes (see S1 Methods). We found no evidence that the presence of *wx1-m7*::*Ac* significantly impacted the overall conclusions drawn from the dataset, nor evidence of epigenetic silencing of GFP transgenes.

## Insertional mutants in the sperm cell-expressed *Zm gex2* cause paternally triggered aberrant seed development

The male-specific transmission defect for the sole affected gene in the Sperm Cell class, Zm00001d005781 (GRMZM2G036832), was notably more severe than the average defect across all *Ds-GFP* mutants identified with decreased transmission through the male (Table 1). This gene is hereafter referred to as *Zm gex2* or *gex2*, for reasons detailed below. The two independent alleles assessed, *gex2-tdsgR82A03* and *gex2-tdsgR84A12*, were associated with transmission rates of 33.4% and 23.1%, respectively. Sequencing confirmed that these *Ds-GFP* elements were inserted into their predicted CDS locations (Fig 7A). In addition to the transmission defect, both alleles, when crossed through the male, conditioned unusual phenotypes: underdeveloped or aborted seeds, as well as ovules with no apparent seed development despite heavy pollination (Fig 7B). These features motivated further investigation of this gene.

Across maize tissues, *gex2* is highly and specifically expressed in sperm cells [38] (Fig 7C). Like many highly expressed genes in mature pollen, it is within 2kb of a transcriptionally active

**Table 2. Characteristics of genes showing Mendelian inheritance.**

| Category | Gene designation (v3) | Gene designation (v4) | Gene Type | Ds-GFP allele | Male transmission rate | Adjusted p-value | Best BLAST Hit, A. thaliana | Predicted Function (B73v4 Gramene) |
|---|---|---|---|---|---|---|---|---|
| Seedling Only | GRMZM2G000052 | Zm00001d002266 | syntelog | tdsgR63F09 | 49.37% | 6.99E-01 | AT2G43020 | Lysine-specific histone demethylase 1 |
| Seedling Only | GRMZM2G111143 | Zm00001d004768 | singleton | tdsgR80E09 | 49.09% | 6.99E-01 | AT5G24318 | Glycosyl hydrolase superfamily protein |
| Seedling Only | GRMZM2G007283 | Zm00001d005036 | syntelog | tdsgR83H05 | 49.22% | 6.99E-01 | AT5G01020 | Serine/threonine-protein kinase |
| Seedling Only | GRMZM2G129209 | Zm00001d007228 | singleton | tdsgR65E02 | 48.00% | 6.83E-01 | AT5G05580 | Omega-3 fatty acid desaturase |
| Seedling Only | GRMZM2G051403 | Zm00001d013295 | singleton | tdsgR46C04 | 51.28% | 6.99E-01 | AT3G55250 | Nuclear pore complex protein Nup214 |
| Seedling Only | AC217975.3_FG001 | Zm00001d022274 | singleton | tdsgR106F04 | 49.37% | 6.99E-01 | AT3G06483 | pyruvate orthophosphate dikinase4 |
| Seedling Only | GRMZM2G100288 | Zm00001d029047 | syntelog | tdsgR76E07 | 49.79% | 8.83E-01 | AT3G51550 | Receptor-like protein kinase FERONIA |
| Seedling Only | GRMZM2G127798 | Zm00001d035925 | singleton | tdsgR53F11 | 48.87% | 6.99E-01 | AT3G02360 | 6-phosphogluconate dehydrogenase1 |
| Seedling Only | GRMZM2G342243 | Zm00001d036283 | syntelog | tdsgR52B09 | 48.45% | 6.99E-01 | AT1G45688 | Late embryogenesis abundant protein group 2 |
| Seedling Only | GRMZM2G044882 | Zm00001d051110 | singleton | tdsgR12H07 | 49.43% | 6.99E-01 | AT5G18430 | GDSL esterase/lipase LTL1 |
| Vegetative Cell | GRMZM5G876898 | Zm00001d002258 | singleton | tdsgR81G05 | 51.26% | 6.80E-01 | AT1G11860 | Aminomethyltransferase |
| Vegetative Cell | GRMZM2G142863 | Zm00001d003947 | syntelog | tdsgR83B04 | 48.73% | 7.03E-01 | AT5G65750 | 2-oxoglutarate dehydrogenase E1 component |
| Vegetative Cell | GRMZM5G827174 | Zm00001d007845 | syntelog | tdsgR52E07 | 47.09% | 2.88E-01 | AT1G10020 | Formin-like protein 18 |
| Vegetative Cell | GRMZM2G045278 | Zm00001d012382 | singleton | tdsgR107C12 | 48.33% | 4.26E-01 | AT3G53990 | Adenine nucleotide alpha hydrolases-like superfamily protein |
| Vegetative Cell | GRMZM2G045278 | Zm00001d012382 | singleton | tdsgR34C11 | 48.78% | 6.80E-01 | AT3G53990 | Adenine nucleotide alpha hydrolases-like superfamily protein |
| Vegetative Cell | GRMZM2G102912 | Zm00001d015242 | singleton | tdsgR99B02 | 49.27% | 9.07E-01 | AT2G24390 | AIG2-like protein |
| Vegetative Cell | GRMZM2G056252 | Zm00001d017840 | syntelog | tdsgR41F01 | 49.62% | 9.36E-01 | AT3G12120 | Delta(12)-fatty-acid desaturase |
| Vegetative Cell | GRMZM5G872068 | Zm00001d017958 | syntelog | tdsgR98H09 | 47.47% | 1.69E-01 | AT1G66200 | Glutamine synthetase root isozyme 3 |
| Vegetative Cell | GRMZM2G136508 | Zm00001d025437 | singleton | tdsgR31H05 | 50.18% | 9.68E-01 | AT2G01170 | Amino acid permease |
| Vegetative Cell | GRMZM2G126858 | Zm00001d026303 | syntelog | tdsgR23D05 | 51.10% | 7.26E-01 | AT1G56145 | Putative leucine-rich repeat receptor-like protein kinase family protein |
| Vegetative Cell | GRMZM2G120136 | Zm00001d026445 | syntelog | tdsgR24D03 | 49.47% | 9.36E-01 | AT1G45180 | E3 ubiquitin-protein ligase MBR2 |
| Vegetative Cell | GRMZM2G006894 | Zm00001d026490 | syntelog | tdsgR02D02 | 49.57% | 9.68E-01 | AT4G30190 | proton-exporting ATPase4 |
| Vegetative Cell | GRMZM2G172751 | Zm00001d027590 | singleton | tdsgR35A08 | 48.21% | 6.80E-01 | AT2G33420 | Protein of unknown function (DUF810 domain) |
| Vegetative Cell | GRMZM2G035243 | Zm00001d027856 | syntelog | tdsgR72D11 | 49.84% | 9.68E-01 | AT1G14330 | Kelch motif family protein |

(*Continued*)

**Table 2.** (Continued)

| Category | Gene designation (v3) | Gene designation (v4) | Gene Type | Ds-GFP allele | Male transmission rate | Adjusted p-value | Best BLAST Hit, A. thaliana | Predicted Function (B73v4 Gramene) |
|---|---|---|---|---|---|---|---|---|
| Vegetative Cell | GRMZM2G016734 | Zm00001d028820 | syntelog | tdsgR27E01 | 48.34% | 6.80E-01 | AT1G56300 | Chaperone DnaJ-domain superfamily protein |
| Vegetative Cell | GRMZM2G142249 | Zm00001d032279 | singleton | tdsgR77F09 | 49.17% | 8.88E-01 | AT3G47730 | ABC2 homolog 15 |
| Vegetative Cell | GRMZM2G114093 | Zm00001d032310 | singleton | tdsgR04G10 | 49.57% | 9.36E-01 | AT5G58950 | Protein kinase superfamily protein |
| Vegetative Cell | GRMZM2G124434 | Zm00001d032950 | singleton | tdsgR01G01 | 48.39% | 6.80E-01 | AT5G28840 | GDP-mannose 35-epimerase |
| Vegetative Cell | GRMZM5G878153 | Zm00001d034799 | singleton | tdsgR103E04 | 49.33% | 9.36E-01 | AT3G03320 | RNA-binding ASCH domain protein |
| Vegetative Cell | GRMZM2G134054 | Zm00001d034839 | singleton | tdsgR32B05 | 50.28% | 9.68E-01 | AT3G07130 | Purple acid phosphatase 15 |
| Vegetative Cell | GRMZM2G134054 | Zm00001d034839 | singleton | tdsgR45E04 | 50.00% | 1.00E+00 | AT3G07130 | Purple acid phosphatase 15 |
| Vegetative Cell | GRMZM2G307402 | Zm00001d036330 | singleton | tdsgR69C04 | 48.26% | 6.80E-01 | AT2G28200 | C2H2-type zinc finger family protein |
| Vegetative Cell | GRMZM2G012263 | Zm00001d037061 | singleton | tdsgR101B03 | 50.08% | 9.80E-01 | AT5G14130 | Peroxidase 64 |
| Vegetative Cell | GRMZM2G012263 | Zm00001d037061 | singleton | tdsgR81E02 | 50.39% | 9.36E-01 | AT5G14130 | Peroxidase 64 |
| Vegetative Cell | GRMZM2G018372 | Zm00001d041514 | syntelog | tdsgR88B08 | 50.55% | 9.36E-01 | AT1G18670 | Protein kinase superfamily protein IBS1-like |
| Vegetative Cell | GRMZM2G095206 | Zm00001d046483 | singleton | tdsgR92A10 | 52.43% | 2.94E-01 | AT4G09750 | NAD(P)-binding Rossmann-fold superfamily protein |
| Vegetative Cell | GRMZM2G089699 | Zm00001d048384 | complex* | tdsgR39B06 | 51.04% | 8.88E-01 | AT1G65680 | beta expansin10a |
| Vegetative Cell | GRMZM5G845021 | Zm00001d048785 | singleton | tdsgR08A07 | 47.30% | 4.25E-01 | AT3G03900 | Adenylyl-sulfate kinase 3 |
| Sperm Cell | GRMZM2G365613 | Zm00001d006218 | singleton | tdsgR60D10 | 51.47% | 7.46E-01 | AT5G42560 | HVA22-like protein i |
| Sperm Cell | GRMZM2G100318 | Zm00001d012128 | syntelog | tdsgR87A03 | 50.03% | 9.79E-01 | AT1G67710 | Putative two-component response regulator family protein |
| Sperm Cell | GRMZM2G172726 | Zm00001d021974 | syntelog | tdsgR35A03 | 51.10% | 7.46E-01 | AT1G19360 | Arabinosyltransferase RRA3 |
| Sperm Cell | GRMZM2G160069 | Zm00001d025834 | singleton | tdsgR31B01 | 47.96% | 7.46E-01 | AT4G16480 | Inositol transporter 4 |
| Sperm Cell | GRMZM2G114899 | Zm00001d034788 | syntelog | tdsgR91F11 | 49.24% | 7.46E-01 | AT1G77280 | Protein kinase protein with adenine nucleotide alpha hydrolases-like domain |
| Sperm Cell | GRMZM2G007659 | Zm00001d042810 | syntelog | tdsgR26G07 | 51.03% | 7.46E-01 | AT2G29680 | Cell division control protein 6 homolog B |
| Sperm Cell | GRMZM2G038252 | Zm00001d043076 | syntelog | tdsgR53C03 | 50.99% | 7.46E-01 | AT4G35550 | WUSCHEL-related homeobox 13a |
| Sperm Cell | GRMZM2G099382 | Zm00001d044109 | singleton | tdsgR106G12 | 49.33% | 7.46E-01 | AT5G47560 | Tonoplast dicarboxylate transporter |
| Sperm Cell | GRMZM2G352898 | Zm00001d048434 | singleton | tdsgR37A04 | 51.31% | 7.46E-01 | AT3G63240 | Type IV inositol polyphosphate 5-phosphatase 7 |

*Beta expansins have proliferated in the maize genome, including by tandem duplication [77]. Thus, this gene cannot be strictly characterized as syntelog; however, the presence of multiple paralogs in the genome indicate it should not be categorized as a singleton gene.

TE, a downstream RLG retrotransposon that displays sperm cell-specific activation (Fig 7C). The *Zm gex2* gene was first identified via EST sequencing of maize sperm cells [44], and subsequently used to isolate the Arabidopsis ortholog, named *GAMETE EXPRESSED2* (*GEX2*) [45].

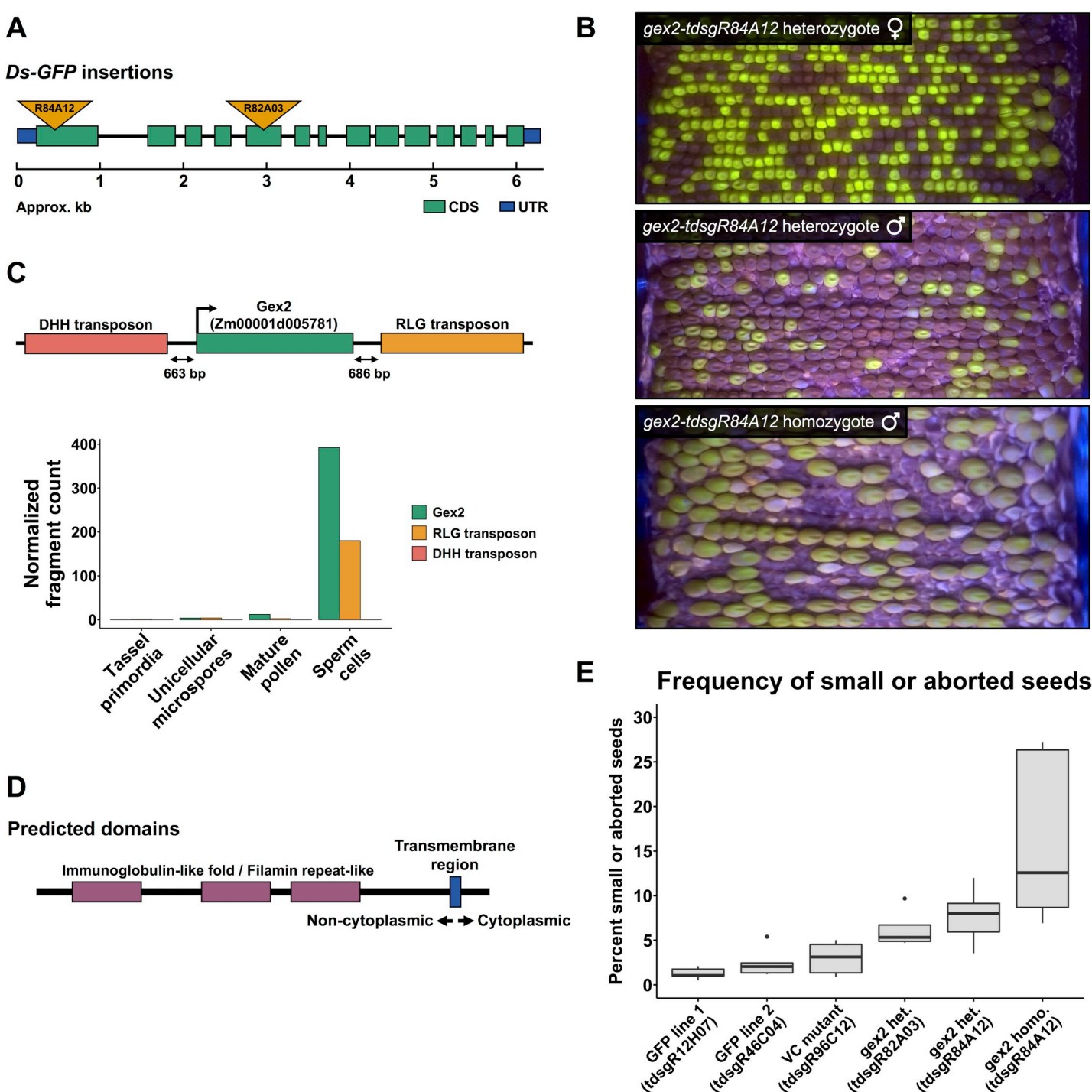

**Fig 7. Mutations in the sperm cell-specific *gex2* gene cause aberrant seed development. (A)** The exon/intron structure of *gex2* (Zm00001d005781/GRMZM2G036832), showing the locations of the two independent *Ds-GFP* insertion mutants. **(B)** Ear projections of *gex2* mutant outcrosses. Top: heterozygote outcrossed as female, showing 1:1 transmission of the GFP-tagged allele. Middle: heterozygote outcrossed as a male, with 26.1% transmission of the mutant allele. Additionally, small seeds and occasional, small gaps between seeds are visible. Bottom: homozygous mutant outcrossed as a male, with many small seeds and large gaps, despite heavy pollination. **(C)** Genomic neighborhood of the GEX2 locus, with two nearby TEs, and their RNA-seq expression levels across male reproductive development. **(D)** Predicted domain structure of Zm GEX2; the amino acid sequence shows 44.2% similarity with Arabidopsis GEX2. **(E)** Quantification of small/aborted seeds resulting from pollination by *gex2* mutant plants and controls. Controls included two *Ds-GFP* lines that did not show transmission defects (*tdsgR12H07* and *tdsgR46C04*) and one *Ds-GFP* line that showed a strong transmission defect in the vegetative cell group (*tdsgR96C12*; 29.5% transmission). A higher percentage of small/aborted seeds was present following pollination by heterozygous *gex2* plants representing the two mutant alleles (*tdsgR82A03* and *tdsgR84A12*), and pollination by homozygous *gex2-tdsgR84A12* plants further increased the percentage of small/aborted seeds.

In Arabidopsis, *GEX2* is necessary for effective double fertilization, causing seed abortion and empty spaces in the silique when a mutant allele is inherited through the male [46]. *Zm gex2* encodes a protein with similar structure and amino acid sequence to its Arabidopsis ortholog (Fig 7D).

Small and aborted seeds were quantified for both *gex2* insertion alleles when outcrossed to wild-type plants (S10 Table, S5A Fig). Two different *Ds-GFP* insertion lines that were not associated with transmission defects (*tdsgR12H07*, *tdsgR46C04*), as well as the *Ds-GFP* associated with the strongest male transmission defect in the Vegetative Cell class (*tdsgR96C12*), were used as controls. Pollination with both *gex2::Ds-GFP* insertion alleles was associated with increased percentages of small or aborted seeds, significantly so in *gex2-tdsgR84A12* (pairwise t-test against *Ds-GFP* controls, all p-values < 0.05), and pollination from *gex2-tdsgR84A12* homozygotes approximately doubled the percentage of aberrant seeds from heterozygotes (pairwise t-test against *Ds-GFP* controls, all p-values < 0.01) (Fig 7E). From the heterozygous *gex2::Ds-GFP* crosses, small seeds with endosperm large enough for DNA preparation were genotyped, and 79.2% were found to harbor the *gex2* mutation, whereas the *tdsgR46C04* control showed Mendelian segregation in small seeds (S5B Fig). These data support the hypothesis that aberrant seed development is induced by fertilization by *gex2::Ds-GFP* sperm.

If the Zm GEX2 protein acts to promote double fertilization similarly to its Arabidopsis ortholog, the arrival of a *gex2::Ds-GFP* pollen tube at the embryo sac could lead to failure of one or both fertilization events. Given an active polytubey block, this could produce the observed gaps between seeds on the ear, resulting from ovules associated with completely failed fertilization, or with very early seed abortion due to single fertilization. Consistent with this possibility, pollination with both heterozygous and homozygous *gex2-tdsgR82A03* alleles resulted in increased seedless area relative to controls (S6 Fig). To test for aberrant fertilization more directly, seed development was assessed at 4 days post-pollination with either wild-type or *gex2-tdsgR84A12* homozygous pollen (Fig 8, Table 3). Typical embryo and endosperm development, as well as indication of the polytubey block (i.e., arrival of only single pollen tubes at the embryo sac), was observed in all ovules assessed from wild-type pollination. In contrast, half of the ovules assessed following pollination with *gex2::Ds-GFP* showed significant evidence of abnormal double fertilization, demonstrating single fertilization of either embryo or endosperm or indication of arrival of more than one pollen tube at the embryo sac (Fisher's exact test, p-value = 0.000241). We conclude that in maize, similarly to Arabidopsis, Zm GEX2 is part of the sperm cell machinery that helps ensure proper double fertilization.

## Discussion

### The *Zm gex2* gene has a conserved role in promoting double fertilization

The generation of a well-replicated developmental time course of transcriptomic data enabled the targeting of highly expressed genes in vegetative and sperm cells for mutational screening. Two independent insertions in the highly and specifically expressed maize sperm cell gene *gex2* led not only to severely reduced transmission through the male, but also, in contrast to other mutations analyzed in this study, to paternally triggered post-fertilization defects. *Zm gex2* was first identified in maize by sperm cell EST sequencing [44], which led to identification of the orthologous gene in Arabidopsis, *GEX2*, and its sperm cell-specific promoter [45]. In Arabidopsis, single fertilization events were observed at increased frequency in *GEX2* mutant-pollinated plants, both for the egg cell and the central cell, leading to an observed increase in aberrant seed development and abortion [46]. Our results in maize are similar, with *gex2* mutant pollen resulting in unfilled ovules, single fertilization events in embryo sacs, and aberrant early seed development from embryo sacs fertilized by *gex2::Ds-GFP* sperm cells. In

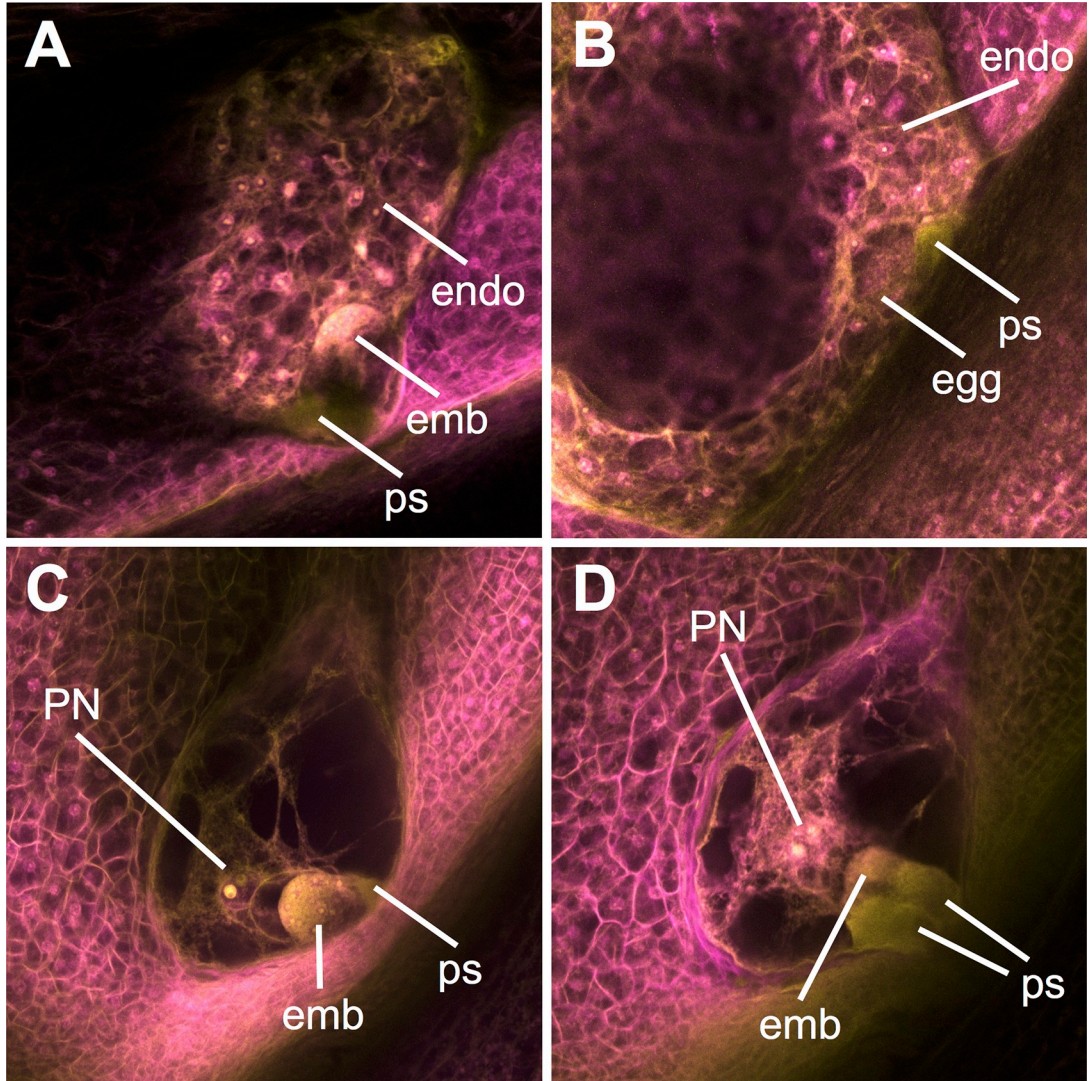

**Fig 8. Pollination by *gex2-tdsgR84A12* leads to aberrant fertilization events and developing seed phenotypes. (A)** Seed development in a typical ovule pollinated by wild-type pollen, with one synergid penetrated by a pollen tube, and both embryo and endosperm development initiated. **(B-D)** Abnormal phenotypes seen following *gex2-tdsgR84A12* pollination. **(B)** Ovule with developing (cellularizing) endosperm but unfertilized egg cell. **(C)** Ovule with developing embryo but unfertilized central cell. **(D)** Ovule with both synergids penetrated by a pollen tube, and a developing embryo and unfertilized central cell. emb = embryo; endo = endosperm; ps = synergid penetrated by a pollen tube; PN = polar nuclei.

Arabidopsis, the interaction between the plasma membrane-localized GEX2 and either the female egg or central cells has been suggested to contribute to gamete attachment. The two orthologues share a predicted domain structure, including a large N-terminal non-cytoplasmic

**Table 3. Seed development at 4 days after pollination by wild-type or *gex2::Ds-GFP* pollen.**

| | One synergid penetrated by a pollen tube | | | Both synergids penetrated by a pollen tube | | |
|---|---|---|---|---|---|---|
| **Pollen parent** | **Both embryo and endosperm** | **Endosperm without embryo** | **Embryo without endosperm** | **Both embryo and endosperm** | **Endosperm without embryo** | **Embryo without endosperm** |
| *gex2-tdsgR84A12/ gex2-tdsgR84A12* | 6 | 2 | 2 | 0 | 0 | 2 |
| Wild-type | 28 | 0 | 0 | 0 | 0 | 0 |

region containing filamin repeat domains potentially contributing to this interaction [46], raising the possibility that Zm GEX2 acts similarly during double fertilization. With conserved *GEX2*-like genes widely distributed throughout the currently sequenced Embryophyta taxa, our results support the idea that, in flowering plants, these genes play a crucial role in double fertilization.

## Maize pollen provides a powerful model for quantifying gene-specific contributions to fitness

The transcriptome dataset also provided a framework to ask broader questions regarding potential relationships between elevated expression and gene function, i.e., utilizing *Ds-GFP* insertions not merely in a genetic screen, but in a mutational interrogation of gene function guided by quantitative hypotheses. Despite the explosion of omic-scale methods to characterize genomes and to measure molecular characters (e.g., transcript levels), our ability to predict phenotypic relevance for specific genes is limited, particularly in multicellular organisms. A simple hypothesis is that a high transcript level at a particular developmental stage implies functional relevance for the associated gene at that stage, and thus potential for phenotypic influence, a hypothesis supported by observations in mice [47]. This study addresses this hypothesis in plants with a systematic assessment of the functional relevance of highly expressed genes in maize pollen, taking advantage of the ease of reciprocal outcross pollination in maize, the availability of a sufficient number of marked and likely null mutations, and the development of an imaging technique that enables sensitive quantitation.

In the post-pollination progamic stage, pollen grains, as independent, genetically distinct organisms, compete to be the first to deliver the sperm cells to the embryo sac for double fertilization. In an outcrossing plant with an extensive stigma and style like maize, there is likely a heightened context for competition among these individuals [10], thus providing a milieu that may be particularly sensitive to genetic perturbation. We reasoned that genes highly expressed in the vegetative cell would tend to contribute to a competitive advantage at this stage, which is responsible for pollen tube germination and growth. Indeed, we found that CDS-insertion alleles for 7 out of 32 (21.9%) tested genes in this class are associated with mild to moderate male-specific transmission defects, with the majority of these defects classified as mild (~45% transmission) and thus detectable only by assessing large populations. In this class, transcript level was significantly correlated with both reduced transmission rate and an increased likelihood of significant non-Mendelian transmission, although the explanatory power is limited ($R^2 < 0.2$), as expected for a complex biological system.

Genetic redundancy could also be predicted to influence the phenotypic outcome of mutating single genes, and there is a suggestive trend consistent with this idea in our dataset (Fisher's exact test, p-value = 0.2117), with 6/7 non-Mendelian alleles classified as singletons in the maize genome (86%), compared to 14/25 singletons in genes harboring Mendelian alleles (56%) (Tables 1 and 2). In addition, the variety of biological processes predicted for genes with documented fitness contributions (Table 1) is consistent with the idea that competitive pollen tube growth requires an array of cellular functions. It should be noted that our approach relies on the availability of *Ds-GFP*-marked insertion alleles, and it seems likely that such availability is biased against genes with severe transmission phenotypes, as these would be selected against in a transposon-mutagenized population. In fact, two large *Mutator* transposon populations show a statistically significant deficit in insertions in genes associated with gametophyte-enriched expression [23]. Notably, both previously described maize genes associated with severe male transmission defect mutants (*rop2*, *apt1*; transmission <15% [11,48]) also would be classified as highly expressed in the vegetative cell ($\log_2(FPKM) > 8$), suggesting that the

trend we found is applicable even to the types of genes most likely to be absent from the *Ds-GFP* insertion population.

Few studies have investigated the relationship between transcriptomic data and quantitative mutant phenotypes, particularly in multicellular organisms. Large-scale screening of the Arabidopsis T-DNA mutant collection for leaf or reproduction-related phenotypes has identified effects in ~4% of lines assessed [49,50], but these efforts were not guided by transcriptome data. Our use of a sensitive phenotypic assay, combined with a focus on sampling mutations in genes that are most highly expressed at a developmental stage relevant to the phenotype tested, seem likely to have contributed to the higher frequency of phenotypic effects we found. In mice, a meta-analysis found a relationship similar to the one we observed, with aberrant phenotypes more likely to be associated with genes highly transcribed in the tissue exhibiting the phenotype [47]. In contrast to these results, genome scale measurement of the fitness costs of gene knockouts via competitive assays in yeast [51,52] and bacteria [53,54] found that, for particular environmental conditions, there was little to no correlation between the expression level of a gene and its impact on fitness in that condition. This could be indicative of differences between the mechanisms underlying single-celled organisms' response to the environment versus those underlying developmental complexity in multicellular organisms. Interestingly, our results are consistent with a recent meta-analysis that identified higher mRNA expression levels as a feature distinguishing gene models with known mutant phenotypes from the overall population of gene models defined by molecular approaches [55].

## Transposable element dynamics in the maize male gametophyte

The transcriptomic time course enabled exploration of the dynamic relationships among developmental progression, gene expression levels, and transcriptional activation of TEs. Understanding TE expression during maize male reproductive development provides an informative comparison to similar analyses in Arabidopsis, an evolutionarily distant plant with a genome landscape that is distinct from maize. Although maize has a higher number and percentage of its genome occupied by TEs compared to Arabidopsis, we found that only a fraction of maize TEs are developmentally dynamic with regards to transcript accumulation. These 'dynamic' TEs tend to be longer elements than average, which suggests that they have protein coding and transpositional potential. From this dynamic TE set, we were able to identify individual elements that are expressed in a number of specific tissues. However, more globally, there is a trend towards activation of TE transcription over the course of the development of the male gametophyte. This finding confirms that both monocots and eudicots have developmental activation of TE expression in pollen. Consistent with our findings, a recent study found that spontaneous retrotransposon mutations are much more frequent through the male than the female in certain maize lines [31]. This conservation suggests that the roles of TE and TE-induced small RNAs during reproductive development may also be conserved between monocots and eudicots [35,56,57].

Although TE activation is conserved in maize and Arabidopsis pollen, we have identified key differences in the timing and location. Maize TE activation is detected earlier (in the unicellular microspore) compared to when it is thought to occur in Arabidopsis [28]. Transcripts from these early-activated TEs in the microspores typically remain detectable through pollen development and in the mature pollen grain, which may be due to continued expression or transcript stability. A second distinction is the location of activation, which in Arabidopsis is confined to the pollen vegetative cell nucleus [28,32], whereas in maize also occurs in sperm cells. *Mutator* family TEs are overrepresented in the pool of sperm-cell transcripts, suggesting that this family of TEs may have evolved (or co-opted) specific regulatory mechanism(s) such as an enhancer element that confers expression in this cell type.

Given our results indicating a linkage between elevated gene expression levels and functional relevance, we also assessed whether similar correlations exist between gene and TE expression locally in the maize genome. We found that in mature pollen and sperm cells there is a positive correlation: the more highly expressed a gene is, the more likely it is to have an up-regulated TE nearby. This tissue-specific correlation is a developmentally-specific co-regulation of gene and TE expression. Notably, there does not appear to be any strong trend linking this co-regulation to gene function. We find instances of local gene/TE coordinate regulation are present in similar proportions in genes with documented transmission defects vs. those showing Mendelian segregation when mutant (14% vs 28% respectively, S5 Table). Therefore, although our data indicate the population of highly expressed maize pollen genes has a tendency to contribute to pollen fitness, and a tendency to be adjacent to pollen-expressed TEs, these two characteristics appear to be independent.

Several potential mechanisms may account for the coordinated gene/TE expression. First, programmed activation of TE expression may influence chromatin, enhancer, or other regulatory functions that influence neighboring genes. Second, the genes and TEs may be directly controlled by the same mechanism of large-scale epigenetic activation, limiting the expression of both to a specific tissue or developmental time point. Third, gene activation may influence the expression of the neighboring TE via read-through transcription. Future studies using alternative transcriptomic approaches (e.g., CAGE or long-read RNA sequencing) will enable the dissection of these possible mechanisms.

## Methods

### Plant materials

Maize inbred line B73 was used for all RNA isolations. Plants were grown in a controlled greenhouse environment (16 hrs light, 8 hrs dark, 80 F day/70 F night) and in the field at the Botany & Plant Pathology Field Lab (Oregon State University, Corvallis, OR) using standard practices. Lines containing *Ds-GFP* insertion alleles were acquired from the Maize Genetics Cooperation Stock Center.

### RNA isolation, library preparation and sequencing

Detailed methods are available in S1 Methods. Briefly, tissue was isolated either by dissection (TP), differential density centrifugation (MS, MS-B and SC), or collection at anthesis (MP). Total RNA from TP, MS, and MP was extracted using a modified Trizol Reagent (Life Technologies) protocol; SC total RNA was extracted via a phenol/chloroform protocol. Poly-A RNA (mRNA) was isolated using streptavidin magnetic beads (New England Biolabs, # S1420S) and a biotin-linked poly-T primer. RNA libraries were prepared and sequenced by the Central Services Lab (CSL) at the Center for Genome Research and Biocomputing (CGRB, Oregon State University) using WaferGen robotic strand specific RNA preparation (WaferGen Biosystems) with an Illumina TruSeq RNA LT (single index) prep kit and run on an Illumina HiSeq 3000 with 100 bp paired-end reads.

### Mapping reads to genes, differential expression assessment and GO enrichment analysis

Ribosomal reads (rRNA) were removed from all samples using STAR, version 2.5.1b [58] to map reads to a repository of maize rRNA sequences (parameters: --outSAMunmapped Within --outSAMattributes NH HI AS NM MD --outSAMstrandFieldintronMotif --limitBAMsortRAM 50000000000 --outReadsUnmapped Fastx). The number of mappable reads generated

from each sample after rRNA removal ranged from approximately 1 million to approximately 41 million, with an average mappable reads of approximately 18 million per sample. Total reads, mapped reads, rRNA contamination, and other statistics are summarized in S1 Table.

rRNA-filtered sequences were mapped to the maize reference genome, version B73 RefGen_v4.33 [36] using STAR, keeping only unique alignments (parameters: --outSAMunmapped Within --outSAMattributes NH HI AS NM MD --outSAMstrandField intronMotif --outFilterMultimapNmax 1 --limitBAMsortRAM 50000000000). Transcript levels of annotated gene isoforms were measured using Cufflinks, version 2.2.1 [59]. FPKM (fragments per kilobase of transcript per million mapped reads) values are shown in S4 Table. Differential expression was calculated between each tissue with Cuffdiff, version 1.0.2, using default parameters. FPKM counts were normalized using the geometric library normalization method. A pooled dispersion method was used by Cuffdiff to model variance. Differential expression results are summarized in S11 Table.

Gene ontology (GO) terms were found for enriched genes in each tissue using the AgriGO 2: GO Analysis Toolkit [60]. Enriched genes were defined as the top 300 significantly differentially expressed genes (q-value) from Cuffdiff output, with ties broken by $log_2$ fold change. Enriched sets were split into up- and down-expressed genes. GO term enrichment was calculated using the singular enrichment analysis method with a Fisher test and Yekutieli multi-test adjustment. GO annotations were based off the maize-GAMER annotation set [61].

## Mapping reads to transposable elements

The rRNA-filtered reads were quality trimmed (QC30) and adapter sequences were removed using BBDUK2 [62]. The remaining sequences were mapped to the whole genome using STAR, allowing mapping to at most 100 'best' matching loci. (parameters: --outMultimapperOrder Random --outSAMmultNmax -1 --outFilterMultimapNmax 100). For paired-end reads, the unmapped reads were re-mapped using single-end approach to maximize the number of mappable reads. The mapping percentage is reported in S3 Table. Because 19% of the total reads in the dataset mapped to more than one location, such reads were mapped to only their best match in the genome, and when multiple best matches existed, they were mapped to all of these loci, and then counted fractionally. For example, if one read maps to 4 TE locations equally well, each TE would receive a weighted value of 0.25 mapped reads. Because the TE expression of the aberrant SC1 biological replicate did not cluster with the other three SC replicates (S1 Fig), it was removed from all subsequent analysis of TE expression.

## Principal component analysis (PCA)

Using the maize gene and TE annotation file available from Ensembl Genomes (v38) [36], a combined annotation file was generated for both genes and TEs to run PCA for all samples. FeatureCounts [63] was used to calculate the accumulation of each gene and TE in all samples following fractional assignment of reads (parameters: -O --largestOverlap -M --fraction -p -C). This counts file was used in DESeq-2 [64] to generate the PCA plot.

## Analysis of transposable elements

The featureCount file (described above) was used as input for differential expression analysis using DESeq2. Since DESeq2 accepts only integers as raw counts, we used 'round' function of R to round the counts to their nearest integers. For differential expression using default parameters for normalization in DESeq2, we only included TEs with a sum total of > = 10 read counts across samples; the rest were categorized as 'not covered' TEs. First, normalized read counts for all TEs were obtained (data in S2 Fig) and then, only TEs (farther than 2kb from

genes) were filtered to be investigated further whereas TEs less than 2kb away from a gene were categorized as 'near genes'.

After selecting seedling as the reference tissue, pairwise volcano plots were generated for all samples against the reference seedling tissue. Each pairwise comparison with the seedling tissue yielded set of TEs with adjusted p-value of either 'NA' or a real number. The set of TEs with a p-value of 'NA' in all pairwise comparisons was added to the count of 'not covered' category since there was not enough statistical power to call differential expression in any of the tissues. The number of TEs statistically significantly up- and down-regulated (adjusted p-value < 0.05) in each tissue was calculated, categorized as 'dynamic TEs' and plotted (Fig 3). Additionally, the number of TEs with adjusted p-value > = 0.05 were categorized as developmentally 'static TEs' as no evidence of TE expression was observed over different developmental time points analyzed. For all categories, the length, family or distance from centromere was calculated based on the published TE annotation file.

### Validation of *Ds-GFP* insertion sites

A FASTA file containing 2 kb of genomic sequence surrounding each *Ds-GFP* insertion site was used as input to a primer3-based tool to generate a pair of specific primers to genotype individual plants from each line (https://vollbrechtlab.gdcb.iastate.edu/tools/primer-server/). The primers used for each *Ds-GFP* line are listed in S6 Table.

To genotype the plants, two 7 mm discs of leaf tissue were collected from each plant using a modified paper punch. The samples were collected in 1.2 ml tubes that fit within a labeled 96 well plate/rack (https://vollbrechtlab.gdcb.iastate.edu/tools/tissue-sample-plate-mapper/) (Phenix Research Products, Candler, NC; M845 and M845BR or equivalent). Genomic DNA was isolated from the leaf punches [65] with the following modifications. An additional centrifugation (3,000 *g* for 10 min.) was added to clear the leaf extracts prior to loading onto a 96-well glass fiber filter plate (Pall, 8032). DNA was eluted from filter plates in 125 μL water, and 2 μL was used as template for PCR. Amplification followed standard PCR conditions using GoTaq Green Master Mix (Promega) with 4% DMSO (v/v) and amplicons were resolved using agarose gel electrophoresis. Lines were genotyped using the pair of *Ds-GFP* line gene-specific primers plus one *Ds*-specific primer (JSR01 GTTCGAAATCGATCGGGATA or JGP3 ACCCGACCGGATCGTATCGG). All lines were also screened by PCR for the presence of *wx1-m7*::*Ac* using primers for *wx1* (CACAGCACGTTGCGGATTTC) and *Ac* (CCGGATCG TATCGGTTTTCG). Followup PCR to test for co-segregation of GFP fluorescence with the presence of the insertion used the appropriate set of three PCR primers (two gene-specific and one *Ds*-specific) and DNA prepared either from endosperm or seedling leaves [66].

### Insertional mutagenesis transmission quantification and statistics

Heterozygous lines with PCR-validated *Ds-GFP* insertion alleles were planted in the Botany & Plant Pathology Field Lab (Oregon State University, Corvallis, OR). All insertions were in coding sequence (CDS) sites. Heterozygous *Ds-GFP* plants were outcrossed to tester plants (*c1/c1 wx1/wx1* or *c1/c1* genetic background) through both the female and the male, with male pollinations made with a heavy pollen load on extended silks (silks that had been allowed to grow for at least two days following cutback). Following harvest, resulting ears were imaged using a custom rotational scanner in the presence of a blue light source and orange filter for GFP seed illumination [43]. Briefly, videos were captured of rotating ears, which were then processed to generate flat cylindrical projections covering the surface of the ear (for examples, see Figs 5 and 7). Seeds were manually counted using the Cell Counter plugin of the Fiji distribution of ImageJ [67]. Ears showing evidence of more than a single *Ds-GFP* insertion (~75% GFP

transmission) were excluded from further analysis. For an allele to be included in the final dataset, we required a minimum of three independent male outcrosses from two different plants.

Seed transmission rates of remaining ears were quantified using a generalized linear model with a logit link function for binomial counts and a quasi-binomial family to correct for over-dispersion between parent lines. By incorporating overdispersion, we allowed for the possibility that seeds on the same ear were not completely independent, and for varying transmission rates between ears associated with a given mutation (e.g. by environmental or maternal effects). A quasi-likelihood approach is more realistic than a simple chi-square test, which assumes that all seeds are independent and transmission rates between ears associated with a given mutation are the same. The dispersion parameter for Sperm Cell and Vegetative Cell categories was approximately 1.8, indicating substantially more heterogeneity among seeds on different ears than is expected in a model which assumes independence. Non-Mendelian inheritance was assessed with a quasi-likelihood test with p-values corrected for multiple testing using the Benjamini-Hochberg procedure to control the false discovery rate at 0.05. Significant non-Mendelian segregation was defined with an adjusted p-value < 0.05. Proportions of genes with male-specific transmission defects in the Seedling, Sperm Cell, and Vegetative Cell categories were compared using a two-sided Fisher's exact test, with significance defined as a p-value < 0.05. A two-sided Fisher's exact test was also used to compare the proportions of male-specific transmission defects in the most highly expressed genes and the less highly expressed in the vegetative cell category. A two-sided test for equality of proportions with continuity correction was used to compare transmission rates in families with partial *Ac* presence. A Git repository containing statistical tests and plotting information for this portion of the study can be found at https://github.com/fowler-lab-osu/maize_gametophyte_transcriptome.

## *Zm gex2* sequence analysis and phenotype characterization

Protein sequence encoded by *Zm gex2* (Zm00001d005781_T002) was retrieved from the Maize Genetics and Genomics Database (MaizeGDB) hosting of the B73 v4 genome [36,68]. Arabidopsis GEX2 protein sequence (AT5G49150.3) was retrieved from the Arabidopsis Information Portal (ARAPORT) Col-0 Araport11 release [69,70]. Maize and Arabidopsis GEX2 protein domains were predicted by InterPro [71], with transmembrane helix predictions by TMHMM [72]. Prediction of land plant species *GEX2* conservation was retrieved from PLAZA, gene family HOM04M006791 [73]. *Zm gex2* gene duplication searches were performed using BLAST [74] and the B73 v4 genome. To confirm the predicted insertion sites for the two *gex2*::*Ds-GFP* alleles, flanking insertion site fragments were PCR-amplified with a gene-specific primer and a *Ds-GFP*-specific primer (DsGFP_3UTR–TGCAAGCTCGAGTT TCTCCA) and sequenced via Sanger sequencing.

To quantify small seed phenotype, mature, dried down maize ears were imaged prior to seed removal from the ear. For small seeds selection, the ear was first visually scanned row by row from the top to the bottom of the ear. Seeds that were noticeably smaller than their surrounding (regular-sized) seeds are carefully removed from the ear using a pin tool. This sometimes required the removal of regular-sized surrounding seeds, which were saved for later counting. A second visual inspection of the ear often resulted in additional small seeds and is recommended. All remaining seeds were then removed from the ear by hand or using a hand corn sheller tool (Seedburo Equip. Co., Chicago, IL). The ear was screened again for any small (flat/tiny) seeds that could have been missed previously. The cob was inspected prior to discarding, and if any small seed was left behind it was removed and accounted for. Small/smaller seeds and regular-sized seeds were counted and counts were recorded (S10 Table). To measure

seedless area, ears were scanned as previously described to create flat surface projections. "Seedless area" was defined as ear surface area that lacked mature or partially developed seeds. Seedless area was quantified as a percentage of total area, as measured with the "Freehand selection" tool of the Fiji distribution of ImageJ [67]. A Git repository containing statistical tests and plotting information for this portion of the study can be found at https://github.com/fowler-lab-osu/maize_gametophyte_transcriptome.

For analysis of embryo sacs by confocal microscopy, tissues were stained with acriflavine, followed by propidium iodide staining [75,76]. After staining, samples were dehydrated in an ethanol series and cleared in methyl salicylate. Samples were visualized on a Leica SP8 point-scanning confocal microscope using excitations of 436 nm and 536 nm and emissions of 540 ± 20 nm and 640 ± 20 nm.

## Supporting information

**S1 Fig. Principal component analysis of gene and transposable element (TE) expression levels.** Two major components, on x- and y-axis, explain 89% of the variance in gene and TE expression levels. Asterisk (*) mark indicates the sample generated as part of this study, whereas other datasets are publicly available. For the sperm cells isolated in this study (SC), the TE expression of one biological replicate did not cluster with the other three (SC1), and therefore was removed from subsequent analyses of expression from TEs. MP-2014, SE, and OV are from [23]; MP-WEB is from [38]; LF is from [37]; MP-LM is from NCBI BioProject 306885 (2015); SC-TD is from [27].
(TIF)

**S2 Fig. Seedling tissue is the appropriate reference for comparison of TE activity. (A)** Steady-state mRNA accumulation of all TEs in different tissues. Datasets generated in this study are marked with an asterisk. **(B)** The number of TEs with zero or near-zero expression levels in different tissues. Seedlings (SE) have the most TEs with low expression levels.
(TIF)

**S3 Fig. Length distribution of categorized TEs subdivided by superfamilies.** Length of TEs in the different TE categories from Fig 2A but further subcategorized by different superfamilies. The violin plots around the box show the kernel probability density of the data. The box represents lower and upper quartile, the line is the median, and the whiskers represent 10–90% range. Red asterisk denotes the mean. 'n' shows the number of TEs in each category for each superfamily.
(TIF)

**S4 Fig. Abundance of TEs near genes in each tissue. (A)** For each tissue type, the top 20,000 expressed genes are distributed along the X-axis in bins of 200, with the highest expressed bin on the far left. The number of TEs near (<2kb) these genes is then counted on the Y-axis (shown in grey bar) and the number of genes with at least 1 TE within 2kb is displayed as black dots. **(B)** Genes filtered for either higher expression in pollen (MP) over sperm cells (SC) (left) or SC>MP (right) were used to determine if the association in Fig 4 is due to sample contamination between SC and MP. Once genes were filtered, the top expressed genes in that tissue were distributed along the X-axis in bins of 200 based on their expression values, with the highest expressed bin on the far left. The number of up- and down-regulated TEs near (<2kb) these genes is then counted on the Y-axis (shown in grey bar) and the number of genes with at least 1 TE within 2kb is displayed as black dots.
(TIF)

**S5 Fig. *gex2* mutant pollen is associated with increased small and aborted seeds in outcross progeny. (A)** Seeds were removed from ears, arranged according to size, and counted. Images of representative seed populations are shown, with the top two rows in each image showing representative fully developed seeds. Rows below the top two contain all of the smaller or aborted seed from that particular ear. **(B)** PCR genotyping of small endosperm seeds from two independent crosses for the two *gex2* alleles show the majority of small seeds harbor the *gex2::Ds-GFP* allele, despite overall reduced transmission of the insertion alleles through the male. Small seeds from control *tsdgR46C04* crosses segregate in a Mendelian fashion.
(TIF)

**S6 Fig. Characterization of *gex2* seedless ear area.** Seedless area was quantified from scanned ear images for *gex2 Ds-GFP* alleles and *Ds-GFP* controls. Pollen from heterozygous *gex2* plants did not show significantly increased seedless area (*gex2-tdsgR82A03* pairwise t-test p-values relative to GFP line 1, GFP line 2, and VC mutant 0.95, 0.96, and 0.74, respectively; *gex2-tds-gR84A12* pairwise t-test p-values 0.19, 0.13, and 0.06, respectively), whereas pollen from homozygous *gex2-tdsgR84A12* plants had significantly increased seedless area (pairwise t-test against *Ds-GFP* controls separately, all p-value < 0.0001).
(TIF)

**S1 Methods. Tissue sample preparation, RNA extraction, and analysis of potential confounding variables in insertional mutagenesis lines.**
(PDF)

**S1 Table. Gene sequencing statistics and availability.** Summary statistics for sequencing data generated in the study.
(XLSX)

**S2 Table. GO term enrichment results.** Differentially expressed genes in developmental categories examined in the study, as well as significantly enriched GO terms associated with these genes.
(XLSX)

**S3 Table. Transposable element sequencing statistics and availability.** Summary statistics and availability for expression datasets used in the analysis of transposable element expression.
(XLSX)

**S4 Table. Genic isoform abundance (FPKM) across developmental stages.** Cufflinks output describing isoform expression by developmental stages, separated by biological replicate.
(TXT)

**S5 Table. Top 20% transcripts by FPKM in Mature Pollen, Sperm Cell and Seedling datasets.** List of top 20% highly expressed genes assigned to the Vegetative Cell, Sperm Cell or Seedling Only classes.
(XLSX)

**S6 Table. Insertional mutagenesis alleles and primers.** List of alleles tested for the presence of *Ds-GFP* insertions by PCR, including primers sequences.
(XLSX)

**S7 Table. Insertional mutagenesis results by line.** Insertional mutagenesis results, separated by line, including marker transmission rates and expression category information.
(XLSX)

**S8 Table. Insertional mutagenesis results by allele.** Insertional mutagenesis results, separated by allele, including marker transmission rates and expression category information.
(XLSX)

**S9 Table. Concordance of seed phenotype with DsGFP genotype.** PCR results from testing *Ds-GFP* presence GFP and non-GFP seeds for selected alleles.
(XLSX)

**S10 Table. *gex2* small seed phenotyping.** *gex2* small seed counting and seedless area results.
(XLSX)

**S11 Table. Differential expression results.** Cuffdiff output comparing expression between tissues examined in this study.
(TXT)

## Acknowledgments

We thank O. Childress, H. Fowler, B. Galardi, B. Hamilton, R. Hartman, and C. Lambert for their tireless seed counting, genotyping, field work, and other technical assistance; and Dr. Lian Zhou for her contributions to maize field genetics. We also thank K. Wimalanathan and T. Shibamoto for computational support at ISU, and M. Dasenko and the Center for Genome Research and Biocomputing for library preparation, sequencing and computational support at OSU. We thank D. Auger for reading the manuscript.

## Author Contributions

**Conceptualization:** Cedar Warman, Kaushik Panda, Erik Vollbrecht, Matthew M. S. Evans, R. Keith Slotkin, John E. Fowler.

**Data curation:** Sam Hokin.

**Formal analysis:** Cedar Warman, Kaushik Panda, Sam Hokin, Duo Jiang, John E. Fowler.

**Funding acquisition:** Erik Vollbrecht, Matthew M. S. Evans, R. Keith Slotkin, John E. Fowler.

**Investigation:** Cedar Warman, Kaushik Panda, Zuzana Vejlupkova, Erica Unger-Wallace, Rex A. Cole, Antony M. Chettoor, Matthew M. S. Evans, John E. Fowler.

**Methodology:** Cedar Warman, Kaushik Panda, Zuzana Vejlupkova, Erica Unger-Wallace, Rex A. Cole, John E. Fowler.

**Project administration:** Erik Vollbrecht, Matthew M. S. Evans, R. Keith Slotkin, John E. Fowler.

**Software:** Cedar Warman, Kaushik Panda, Sam Hokin.

**Supervision:** Erik Vollbrecht, Matthew M. S. Evans, R. Keith Slotkin, John E. Fowler.

**Visualization:** Cedar Warman, Kaushik Panda, Zuzana Vejlupkova, Sam Hokin, Matthew M. S. Evans, John E. Fowler.

**Writing – original draft:** Cedar Warman, Kaushik Panda, Zuzana Vejlupkova, Sam Hokin, Erica Unger-Wallace, R. Keith Slotkin, John E. Fowler.

**Writing – review & editing:** Cedar Warman, Kaushik Panda, Zuzana Vejlupkova, Sam Hokin, Erica Unger-Wallace, Duo Jiang, Erik Vollbrecht, Matthew M. S. Evans, R. Keith Slotkin, John E. Fowler.

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
