## [Decision Letter · Decision Letter 0]

10 Dec 2019

Dear Dr Fowler,

Thank you very much for submitting your Research Article entitled 'Highly expressed maize pollen genes display coordinated expression with neighboring transposable elements and contribute to pollen fitness' to PLOS Genetics. Your manuscript was fully evaluated at the editorial level and by independent peer reviewers. The reviewers appreciated the attention to an important problem, but raised some substantial concerns about the current manuscript. Based on the reviews, we will not be able to accept this version of the manuscript, but we would be willing to review again a much-revised version. We cannot, of course, promise publication at that time.

There are several very interesting observations presented in this manuscript.  Several of the reviewers (and I) noted that the paper is a bit difficult to follow and summarize as it really is two distinct studies that are not all that well linked.  You should consider how to better link these studies of TEs/expression and fitness as they are both interesting but are difficult to push together in a cohesive narrative.  In addition, the reviewers pointed out a number of specific issues that will need to be addressed either through providing more clarity of altering some analyses.  

If you decide to revise the manuscript for further consideration at PLOS Genetics, please aim to resubmit within the next 60 days, unless it will take extra time to address the concerns of the reviewers, in which case we would appreciate an expected resubmission date by email to plosgenetics@plos.org.

[LINK]

We are sorry that we cannot be more positive about your manuscript at this stage. Please do not hesitate to contact us if you have any concerns or questions.

Yours sincerely,

Nathan M. Springer

Associate Editor

PLOS Genetics

Gregory P. Copenhaver

Editor-in-Chief

PLOS Genetics

Reviewer's Responses to Questions

**Comments to the Authors:**

Reviewer #1: I think the title is problematic, because of the word “and”. The paper has two parts: in the first they demonstrate TE expression in male development similar to what was shown for Arabidopsis, but find key differences in timing and location. No connection between TE part of the paper and the transmission defects part of the paper; could be a coincidence, nothing more. Much of work on the GEX2 mutant is duplicative of what was already published in Arabidopsis. Transmission defects for vegetative-expressed genes are modest, except for one –surprised they found anything, as many pollen proteins are members of multi-gene families (all expressed in pollen). So gene redundancy might be expected to cover for any defect in another. They should provide info. about possible gene redundancy to explain lack of/presence of phenotypes for the genes they chose to test. I am not convinced by their idea that highly expressed genes are likely to be more important for function. Paper is not user-friendly – e.g., if a reader is interested in knowing what genes were tested and which ones did not show transmission defects, the supplemental tables don’t help, long gene identifier numbers, nothing else. A table of annotations for all tested genes should be in the body of the paper.

Line 2: strange to say that the male gametophyte is essential for subsequent initiation of seed development – with such an argument, it must be important for every subsequent step in the life cycle. Just say pollen tube growth and sperm delivery.

Line 45: insert the word the in front of the word pollen

Lines 50-52: a role for GEX2 already demonstrated in Arabidopsis (reference 47); should be mentioned here. Similarly, in line 69- role for GEX2 in fertilization already established – if the authors insist on this phrasing, they need to say maize gex2 throughout. Similarly in lines 138-140.

Line 61; what does “this process” refer to? Pollen tube growth?

Line 240 – the word somatic is not needed

Line 272, vice, not vise

Line 283-285, they should also admit “or neither”, i.e. expression levels of TE and genes could be a coincidence, neither affecting the other.

Line 289, there is no such thing as high or low tissue-specificity, either something is specific or it is not. Otherwise use the word enriched.

Line 307 – competitive delivery of gametes is unusual phrasing, the competition is at the level of pollen tube growth.

line 356-362. I don’t understand the math – 6 of 7 and then 6 of 12.

Line 394, cite ref. 46 after EST sequencing, and the reference to the name GEX2 (ref 50), is missing. Neither ref should be at the end of the sentence, as “no mutant has been described” is not mentioned in either ref. 46 or ref 50. The statements on lines 495-497 are better and accurate.

Line 400 “has been described” is passive voice – Ref. 47 shows it.

Line 401 – of course it encoded a protein similar to Arabidopsis Gex2, the way Gex2 was identified was by using maize ESTs (ref. 46) to find the Arabidopsis homolog (ref 50).

Lines 405-441 essentially are showing in maize what was already shown (ref 47) in Arabidopsis. Probably does not merit this much space.

Line 507-508 Why mention “we cannot rule out….”

Lines 512-520 – why discuss these other mutants? They don’t bear on the work in this paper.

Lines 555 mild to moderate is an overstatement here, 5 were mild, only one was moderate.

Line 560 – delete extrapolation comment, it is pointless from such a small sample.

Discussion is much too long, could be cut by 40%.

Reviewer #2: In this manuscript by Warman et al., the authors performed RNA-seq at different stages of pollen development to analyze the expression patterns of both genes and transposable elements and then screened mutants in highly expressed genes for transmission defects that imply functional importance of these genes. The highlight of this manuscript was the use of GFP-marked insertion mutants followed by clever imaging to identify genes that reduce pollen fitness. The authors then characterized the aborted seed phenotype seen in paternally inherited gex2 mutants, showing that this gene is involved in double fertilization.

I do have some concerns about this manuscript, specifically with the analysis of the transcriptome data and the integration of transcriptome data with the analysis of transmission defects. For example, the title is misleading. The authors show that several highly expressed maize pollen genes contribute to pollen fitness, but as far as I could tell, only one of these functionally tested genes is near a TE with coordinated expression. Furthermore, the result that highly expressed genes are enriched for up-regulated TEs nearby is interesting but is not well supported by the data presented. What I could not determine from the figures is how many genes have at least one up-regulated TE within 2 kb or how TEs that overlap genes are treated. In Figure 4A, the y-axis shows the number of TEs within 2 kb of bins of 200 genes, but since a single gene may have multiple up-regulated TEs within that vicinity, it is not clear how many genes are associated with up-regulated TEs. With respect to overlap, the authors acknowledge earlier in the manuscript that TEs within 2 kb of a gene introduce complications due to overlaps and the potential for read-through transcription, so it is not clear why this major conclusion of the paper is not more thoroughly assessed to rule out this possibility.

The conclusion that TE transcription in maize is largely static across development is surprising and contradicts prior studies in maize using EST or RNA-seq data, but this difference is not addressed in the manuscript. In looking through the methods, I am wondering if the number of “static” transcripts might be exaggerated by not using an expression cutoff to filter out lowly expressed transcripts that lack power to be called DE and by partially counting ambiguous reads to multiple locations. Although I am not entirely sure from the methods how this analysis was performed since DE TEs were defined using DESeq2, which only allows integer counts, while FeatureCounts is set to output fractions for multi-mapped reads. The methods should also describe how read counts were normalized and what parameters were used to call significantly DE features.

Finally, in Figure 2B the authors conclude that dynamic TEs tend to be longer than all TEs. However, it is not clear from this plot how much of this difference in length distribution results from the difference in TE types in the dynamic set. Since LTR Unknown TEs are on average larger than DNA transposons, the increased presence of LTRs alone could explain the pattern. If the authors want to conclude something about TE lengths in the different sets, TE lengths should be compared within each TE type.

Reviewer #3: This was a fascinating and clearly written paper. The authors describe an RNA-seq based analysis to confirm that the derepression of transposons in male gametophytes previously reported in arabidopsis appears to be a more general property of angiosperms. They demonstrate the use of a novel screen to quantitatively estimate the effect of KO mutants on pollen fitness in a high throughput fashion and use this screen to qualify fitness effects for 30 genes. There is suggestive but not entirely conclusive evidence that genes which are highly expressed in the pollen vegetative cell are more likely to play rolls in determining pollen fitness. Finally the authors identify and characterize a novel mutant that appears to play a role in controlling the odds of proper double fertilization of both the central cell and egg cell of megagametophytes. The last is an area of growing interest from both a flowering plant evo/devo perspective as well as being an area of economic impact and applied research interest.

As I think is clear from my summary I found this to be a very strong manuscript that will be of interest to a broad audience. I do, however, have a couple of minor concerns/questions:

1) Around like 343 the authors mention that they tested for changes in transmission rate using "h quasi-likelihood tests on generalized linear models with a logit link function for binomial counts." It would be good to include a little discussion of why this test was employed rather than a simple chi-square test.

2) For Figure 2B and 2D, violin plots may be more informative than simple box plots.

3) Unless I missed it, it doesn't seem like there is any discussion of the fact that genes with significant transmission defects, even only in the male direction, are likely underpresented in most reverse genetics populations. I would strongly suggest touching on this point in the discussion. If the data are available, it would also be quick and interesting to check whether the genes the authors identify has highly expressed in pollen vegetative cell genes indeed over or under represented in the reverse genetics population employed by the authors (relative to genes highly expressed in seedings or pollen sperm nuclei). But the test is a nice to have, not a must have.

**Have all data underlying the figures and results presented in the manuscript been provided?**

Reviewer #1: Yes

Reviewer #2: None

Reviewer #3: Yes

PLOS authors have the option to publish the peer review history of their article (what does this mean?). If published, this will include your full peer review and any attached files.

Reviewer #1: No

Reviewer #2: No

Reviewer #3: No

---

## [Decision Letter · Decision Letter 1]

17 Feb 2020

Dear Dr Fowler,

Thank you very much for submitting your Research Article entitled 'High expression in maize pollen correlates with genetic contributions to pollen fitness as well as with coordinated transcription from neighboring transposable elements' to PLOS Genetics. Your manuscript was fully evaluated at the editorial level and by independent peer reviewers. The reviewers appreciated the attention to an important topic but identified just a couple minor aspects of the manuscript that should be improved.

Two reviewers were satisfied with the revisions but one reviewer noted some minor issues.  In addition, I think there may be one other detail that should be added relative to your analysis of the transmission phenotypes.  I did not find the specific location of the insertion sites (5' UTR, coding, 3' UTR).  I might have missed this but if this is not present could you please add this.  This will help to confirm that there are similar frequencies of coding insertions and also will reveal if coding/UTR insertions show any differences in frequency of transmission defects. 

We therefore ask you to modify the manuscript according to the review recommendations before we can consider your manuscript for acceptance. Your revisions should address the specific points made by each reviewer.

We hope to receive your revised manuscript within the next 30 days. We do not anticipate a need to send this back out for another evaluation by external reviewers.  If you anticipate any delay in its return, we would ask you to let us know the expected resubmission date by email to plosgenetics@plos.org.

[LINK]

Yours sincerely,

Nathan M. Springer

Associate Editor

PLOS Genetics

Gregory P. Copenhaver

Editor-in-Chief

PLOS Genetics

Reviewer's Responses to Questions

**Comments to the Authors:**

Reviewer #1: The authors have sufficiently addressed most of my comments.

On line 525, I don't see the necessity to state "To our knowledge, this is the largest yet...." In the same way that I don't like claims of first. I suggest removing this phrasing: let the paper stand on its merits, without claims of size, priority, etc.

line 584, they state here that the two points in the original title are independent, although this is somewhat burying the lede. In this section of the discussion they use the word "tendency" whereas the title, using the word correlates, sounds more definitive - so I think the title could be further revised to reflect reality, as stated in the discussion, i.e. tendencies, and independent.

In their response to reviews they state: "In addition, we note

that the maize and Arabidopsis nomenclature conventions also distinguish the genes, as maize uses

all lower case letters (gex2), whereas Arabidopsis uses all upper case (GEX2)" The average reader will not be familiar with the nomenclature rules of both maize and Arabidopsis (lower case, upper case), so it is important (for readability/understanding) to distinguish them throughout.

Reviewer #2: I appreciate the changes that the authors made to clarify the separate findings related to the transcriptome analyses and transmission defect / phenotypic characterizations. My technical concerns with the manuscript have been appropriately addressed and the authors have added sufficient clarifications on methods where needed.

Reviewer #3: The authors have done an excellent job of addressing my concerns from the first round of review.

**Have all data underlying the figures and results presented in the manuscript been provided?**

Reviewer #1: Yes

Reviewer #2: Yes

Reviewer #3: None

PLOS authors have the option to publish the peer review history of their article (what does this mean?). If published, this will include your full peer review and any attached files.

Reviewer #1: No

Reviewer #2: No

Reviewer #3: No

---

## [Editor Report · Decision Letter 2]

27 Feb 2020

Dear Dr Fowler,

We are pleased to inform you that your manuscript entitled "High expression in maize pollen correlates with genetic contributions to pollen fitness as well as with coordinated transcription from neighboring transposable elements" has been editorially accepted for publication in PLOS Genetics. Congratulations!

Yours sincerely,

Nathan M. Springer

Associate Editor

PLOS Genetics

Gregory P. Copenhaver

Editor-in-Chief

PLOS Genetics

Comments from the reviewers (if applicable):

**Data Deposition**

http://datadryad.org/submit?journalID=pgenetics&manu=PGENETICS-D-19-01653R2

Press Queries

---

## [Editor Report · Acceptance letter]

13 Mar 2020

PGENETICS-D-19-01653R2 

High expression in maize pollen correlates with genetic contributions to pollen fitness as well as with coordinated transcription from neighboring transposable elements 

Dear Dr Fowler, 

We are pleased to inform you that your manuscript entitled "High expression in maize pollen correlates with genetic contributions to pollen fitness as well as with coordinated transcription from neighboring transposable elements" has been formally accepted for publication in PLOS Genetics! Your manuscript is now with our production department and you will be notified of the publication date in due course.

With kind regards,

Jason Norris

PLOS Genetics

On behalf of:
